# Expected Variational Inequalities

**Brian Hu Zhang** [* 1]  **Ioannis Anagnostides** [* 1]  **Emanuel Tewolde** [1 2]  **Ratip Emin Berker** [1 2]  **Gabriele Farina** [3]
**Vincent Conitzer** [1 2 4]  **Tuomas Sandholm** [1 5 6 7]

## Abstract

Variational inequalities (VIs) encompass many fundamental problems in diverse areas ranging from engineering to economics and machine learning. However, their considerable expressivity comes at the cost of computational intractability. In this paper, we introduce and analyze a natural relaxation—which we refer to as *expected variational inequalities (EVIs)*—where the goal is to find a *distribution* that satisfies the VI constraint *in expectation*. By adapting recent techniques from game theory, we show that, unlike VIs, EVIs can be solved in polynomial time under general (nonmonotone) operators. EVIs capture the seminal notion of *correlated equilibria*, but enjoy a greater reach beyond games. We also employ our framework to capture and generalize several existing disparate results, including from settings such as smooth games, and games with coupled constraints or nonconcave utilities.

## 1. Introduction

*Variational inequalities (VIs)* provide a unifying framework for analyzing a wide range of optimization and equilibrium problems. They have a host of important applications in engineering and economics (Facchinei & Pang, 2003), including identifying stationary points in constrained optimization; computing Nash equilibria in multi-player games (Nash, 1951), such as Cournot's classical model of oligopoly (Cournot, 1838); predicting economic activity—commodity prices and consumer consumption—in a closed, competitive economy (Arrow & Debreu, 1954), which is at the heart of general equilibrium theory; traffic equilibrium

problems—estimating the steady-state of a congested network wherein users compete for its resources (Dafermos, 1980); frictional contact problems in mechanical engineering (Capatina, 2014); and pricing options, a foundational problem in financial economics (Black & Scholes, 1973).

Formally, in a general form, a VI can be defined as follows.[1]

**Definition 1.1.** Let $\mathcal{X}$ be a convex and compact subset of $\mathbb{R}^d$ and $F : \mathcal{X} \to \mathbb{R}^d$ a bounded map. The *variational inequality (VI)* problem asks for a point $x \in \mathcal{X}$ such that

$$\langle F(\boldsymbol{x}), \boldsymbol{x}' - \boldsymbol{x} \rangle \geq 0 \quad \forall \boldsymbol{x}' \in \mathcal{X}. \tag{1}$$

For computational purposes, it is common to consider the $\epsilon$-*approximate* VI problem, wherein the right-hand side of (1) is replaced by $-\epsilon$ for some precision parameter $\epsilon > 0$.

Definition 1.1 abstracts the description of $F$ and $\mathcal{X}$. As a concrete example, when $F : \boldsymbol{x} \mapsto -\nabla u(\boldsymbol{x})$ is the negative gradient of a differentiable function $u : \mathcal{X} \to \mathbb{R}$, the solutions to (1) are points that satisfy the first-order optimality conditions for maximizing $u$ (Boyd & Vandenberghe, 2004).

Unfortunately, the considerable expressivity of VIs comes at the expense of *intractability*: even when $F$ is linear and $\epsilon$ is an absolute constant, identifying an $\epsilon$-approximate VI solution is computationally hard; this follows readily from the intractability of Nash equilibria—under plausible complexity assumptions (Daskalakis et al., 2008; Chen et al., 2009; Rubinstein, 2016). Unconditional, query-complexity lower bounds have also been established (Hirsch et al., 1989; Babichenko, 2016); *cf.* Milionis et al. (2023) and Hart & Mas-Colell (2003) for other pertinent impossibility results.

This bleak realization has shifted the focus of contemporary research primarily to characterizing specific subclasses of VIs that elude those complexity barriers, with the ensuing line of work flourishing in recent years. Some notable examples include the classical *Minty property* (Facchinei & Pang, 2003; Mertikopoulos & Zhou, 2019), as well as certain relaxations thereof (Diakonikolas et al., 2021; Böhm, 2023; Bauschke et al., 2021; Combettes & Pennanen, 2004; Gorbunov et al., 2023; Cai et al., 2024b; Alacaoglu et al., 2023;

---

[*]Equal contribution [1]Carnegie Mellon University [2]Foundations of Cooperative AI Lab (FOCAL) [3]Massachusetts Institute of Technology [4]University of Oxford [5]Strategy Robot, Inc. [6]Strategic Machine, Inc. [7]Optimized Markets, Inc. Correspondence to: Brian Hu Zhang <bhzhang@cs.cmu.edu>, Ioannis Anagnostides <ianagnos@cs.cmu.edu>.

*Proceedings of the 42$^{nd}$ International Conference on Machine Learning*, Vancouver, Canada. PMLR 267, 2025. Copyright 2025 by the author(s).

---

[1]As is standard, a VI throughout this paper refers to the *Stampacchia* VI *(SVI)* (Kinderlehrer & Stampacchia, 2000).

Pethick et al., 2022; Lee & Kim, 2021; Patris & Panageas, 2024; Choudhury et al., 2024; Anagnostides et al., 2024).

Despite these important advances, the scope of such results is severely restricted. In this paper, we pursue a different, orthogonal avenue. Instead of restricting the class of *problems* to achieve computational tractability, we relax the underlying *solution concept*. Our main research question is:

> *Are there meaningful relaxations of the VI problem that can always be solved efficiently?*

When specialized to games, this question can be seen as part of the research agenda recently outlined by Daskalakis (2022) in his address at the Nobel symposium about equilibrium computation in *nonconcave games*—a major, new frontier in the interface of game theory and optimization.

## 1.1. Our Contribution: The Expected VI Problem

To make progress on that central question, we introduce a natural relaxation of VIs (in the context of Definition 1.1).

**Definition 1.2.** Given a set of *deviations* $\Phi \subseteq \mathcal{X}^{\mathcal{X}}$, the $\epsilon$-*approximate* $\Phi$-*expected variational inequality ($\Phi$-EVI)* problem asks for a distribution $\mu \in \Delta(\mathcal{X})$ such that

$$\mathbb{E}_{\boldsymbol{x} \sim \mu} \langle F(\boldsymbol{x}), \phi(\boldsymbol{x}) - \boldsymbol{x} \rangle \geq -\epsilon \quad \forall \phi \in \Phi.$$

(The above definition does not specify how $\mathcal{X}, F, \Phi$, and $\mu$ should be represented for computational purposes, but we will be explicit about representation whenever it is relevant.)

In words, Definition 1.2 only imposes (approximate) non-negativity *in expectation* for points $\boldsymbol{x}$ drawn from $\mu \in \Delta(\mathcal{X})$. It certainly relaxes Definition 1.1: if $\boldsymbol{x}$ satisfies (1), then the distribution $\mu$ that always outputs $\boldsymbol{x}$ is also a $\Phi$-EVI solution. $\Phi$-EVIs are thus no harder than VIs (assuming that solutions exist). However, as we shall see, the primary justification of $\Phi$-EVIs is that they can be easier than VIs.

Definition 1.2 is crucially parameterized by $\Phi$; the larger the set of deviations $\Phi$, the tighter the set of solutions. As will become clear, Definition 1.2 is intimately connected with notions of *correlated equilibrium (CE)* from game theory (*e.g.*, Aumann, 1974). The more permissive case where $\Phi$ comprises only constant functions, $\Phi = \Phi_{\mathsf{CON}} = \{\phi_{\boldsymbol{x}} : \boldsymbol{x} \in \mathcal{X}\}$ where $\phi_{\boldsymbol{x}}(\boldsymbol{x}') = \boldsymbol{x}$ for all $\boldsymbol{x}' \in \mathcal{X}$, is perhaps the most basic relaxation of Definition 1.1; we call the $\Phi_{\mathsf{CON}}$-EVI problem simply the *EVI problem*.

**Algorithms and complexity for $\Phi$-EVIs** As it turns out, imposing no constraints on $\Phi$ results in an impasse: $\Phi$-EVIs are in general tantamount to regular VIs—thereby being PPAD-hard (Corollaries 3.7 and 3.9). On the other hand, unlike general VIs, one of our key contributions is to show

that when $\Phi$ contains only linear maps, $\Phi_{\mathsf{LIN}}$, $\Phi$-EVIs can be solved in time polynomial in the dimension $d$ and $\log(1/\epsilon)$ (Theorem 4.1), establishing the promised computational property that separates EVIs from VIs. This result is based on *ellipsoid against hope (EAH)*, the seminal algorithm of Papadimitriou & Roughgarden (2008) developed for computing correlated equilibria in multi-player games. (Section 2.1 gives a self-contained overview of EAH.) In doing so, we extend the scope of that algorithm to a much broader class of problems well beyond the realm of game theory. Notably, Theorem 4.1 applies even when $\mathcal{X}$ is given implicitly through a membership oracle; this extension makes use of the recent technical approach of Daskalakis et al. (2025), discussed in more detail in Section 4.

One limitation of Theorem 4.1 is that it relies on the EAH algorithm, which is slow in practice. We address this by also establishing more scalable algorithms that use convex quadratic optimization (Theorem 4.3) instead of the ellipsoid algorithm. As a byproduct, we obtain the best-known algorithm for linear-swap regret minimization over explicitly represented polytopes, improving on Daskalakis et al. (2025) by reducing the per-iteration complexity.

In addition to their more favorable computational properties, we further show that $\Phi$-EVIs admit (approximate) solutions under more general conditions than their associated VIs—namely, without $F$ being continuous (Theorem 3.1); Section 3 documents further interesting aspects on existence.

**Connection to other solution concepts** As we have alluded to, $\Phi$-EVIs generalize (Examples 5.1 and 5.2) the seminal concept of a *(coarse) correlated equilibrium à la* Aumann (1974) and Moulin & Vial (1978) in finite games, and more generally $\Phi$-*equilibria* (Greenwald & Jafari, 2003; Stoltz & Lugosi, 2007; Gordon et al., 2008) of concave games. What is more surprising is that $\Phi_{\mathsf{LIN}}$-EVIs *refine* CEs even in normal-form games; we give illustrative examples, together with an interpretation, in Section 5. We also note that $\Phi$-EVIs can be used even in games with nonconcave utilities (Daskalakis, 2022; Cai et al., 2024a) or noncontinuous gradients (as in nonsmooth optimization), as well as in (pseudo-)games with *coupled constraints* (*cf.* Bernasconi et al., 2023 and Appendix A for related work).

**Further properties** As further motivation, we show that for certain structured problems, such as *(quasar-)concave* optimization and *polymatrix* zero-sum games, EVIs essentially coincide with VIs (Propositions 6.1 and 6.3).

Finally, in certain applications, one might be interested in a VI solution mainly insofar as it provides guarantees in terms of an underlying objective, such as misclassification error or social welfare. Through that prism, the question is whether performance guarantees for VIs can be translated

| Result | Description | Reference |
|---|---|---|
| Existence of ($\epsilon$-approx.) solutions | Under Lipschitz cont. for $\Phi$ and bounded $F$ | Theorem 3.1 |
| Complexity with nonlinear $\Phi$ | PPAD-hardness with linear $F$ and $\epsilon = \Theta(1)$ | Corollaries 3.7 and 3.9 |
| Algorithms for linear $\Phi$ | • $\mathrm{poly}(d, \log(1/\epsilon))$-time via EAH 
 • $\mathrm{poly}(d, 1/\epsilon)$-time via $\Phi$-regret minimization | Theorem 4.1 
 Theorem 4.3 |
| Equivalence between VIs-EVIs | • Quasar-concave functions (Definition 6.2) 
 • $\boldsymbol{x} \mapsto \langle F(\boldsymbol{x}), \boldsymbol{x}' - \boldsymbol{x} \rangle$ concave for all $\boldsymbol{x}'$ | Proposition 6.3 
 Proposition 6.1 |
| Performance guarantees for EVIs | Under smoothness (Definition 7.1) | Theorem 7.4 |

*Table 1.* Our main results concerning $\Phi$-EVIs (Definition 1.2).

to EVIs as well. In Section 7, we establish a framework for accomplishing that (Definition 7.1) by extending the celebrated *smoothness* framework of Roughgarden (2015), and provide interesting examples beyond game theory.

Taken together, these properties provide compelling justification for $\Phi$-EVIs as a solution concept in place of VIs. Table 1 gathers our main results. (Proofs are in Appendix C.)

## 2. Preliminaries

This section provides some basic notation and background together with an overview of the EAH algorithm. Additional preliminaries, which are not necessary for the main body, are given later in Appendix B.

**Notation** We use boldface, lowercase letters, such as $\boldsymbol{x}$ and $\boldsymbol{y}$, to denote vectors in a Euclidean space. Capital, boldface letters, such as $\mathbf{A}$, represent matrices. For $\boldsymbol{x}, \boldsymbol{x}' \in \mathbb{R}^d$, we use $\langle \boldsymbol{x}, \boldsymbol{x}' \rangle$ to denote their inner product. $\|\boldsymbol{x}\| := \sqrt{\langle \boldsymbol{x}, \boldsymbol{x} \rangle}$ is the Euclidean norm of $\boldsymbol{x}$. $\mathcal{B}_r(\boldsymbol{x})$ is the (closed) Euclidean ball centered at $\boldsymbol{x}$ with radius $r > 0$. $\mathrm{conv}(\cdot)$ represents the convex hull. An *endomorphism* on $\mathcal{X}$ is a function mapping $\mathcal{X}$ to $\mathcal{X}$.

Returning to Definition 1.2, for computational purposes, we assume throughout that $F$ has an explicit polynomial representation, so that $F(\boldsymbol{x}) \in \mathbb{R}^d$ can be evaluated in $\mathrm{poly}(d)$ time. Further, there exists $B > 0$ such that $\|F(\boldsymbol{x})\| \leq B$ for all $\boldsymbol{x} \in \mathcal{X}$. We will also restrict the support $\mathrm{supp}(\mu)$ of $\mu$ to be $\mathrm{poly}(d, 1/\epsilon)$, unless stated otherwise. With regard to $\mathcal{X}$, we assume that we have *oracle access*. In particular, we consider the following three types of oracle access.

- *Membership*: given $\boldsymbol{x} \in \mathbb{R}^d$, decide whether $\boldsymbol{x} \in \mathcal{X}$.
- *Separation*: given $\boldsymbol{x} \in \mathbb{R}^d$, decide whether $\boldsymbol{x} \in \mathcal{X}$; if not, return a hyperplane that *separates* $\boldsymbol{x}$ from $\mathcal{X}$.
- *Linear optimization*: Given $\boldsymbol{u} \in \mathbb{R}^d$, return a vector in $\arg\max_{\boldsymbol{x} \in \mathcal{X}} \langle \boldsymbol{x}, \boldsymbol{u} \rangle$.

In addition, we will assume that $\mathcal{X} \subseteq \mathcal{B}_R(\mathbf{0})$ for some $R \leq \mathrm{poly}(d)$, and $\mathcal{X}$ contains a ball of radius 1 in its relative

interior; this is a standard regularity condition that ensures $\mathcal{X}$ is geometrically well-behaved, which can be met by bringing $\mathcal{X}$ into isotropic position (Appendix B). Under this assumption, the three oracles listed above are polynomially equivalent (Grötschel et al., 1993; 1981). As a result, we may assume that $\mathcal{X}$ is given implicitly via a $(\mathrm{poly}(d))$-time membership oracle, which suffices for Theorem 4.1.

All our positive results with respect to the set of linear endomorphisms $\Phi_{\mathsf{LIN}}$ readily carry over to affine endomorphisms as well.

### 2.1. Ellipsoid Against Hope

This *ellipsoid against hope (*EAH*)* algorithm was famously introduced by Papadimitriou & Roughgarden (2008) to compute correlated equilibria in multi-player games. We proceed with an overview of EAH, and in particular a generalized version thereof, crystallized by Farina & Pipis (2024).

Consider an arbitrary optimization problem of the form

$$\text{find} \quad \mu \in \Delta(\mathcal{X}) \quad \text{s.t.} \quad \mathbb{E}_{\boldsymbol{x} \sim \mu} \langle \boldsymbol{y}, G(\boldsymbol{x}) \rangle \geq 0 \,\forall \boldsymbol{y} \in \mathcal{Y}, \tag{2}$$

where $\mathcal{Y} \subseteq \mathbb{R}^m$, and $G : \mathcal{X} \to \mathbb{R}^m$ is an arbitrary function. Suppose that we are given an evaluation oracle for $G$ and a separation oracle for $\mathcal{Y}$. Assume further that we are given a *good-enough-response (*GER*)* oracle, which, given any $\boldsymbol{y} \in \mathcal{Y}$, returns $\boldsymbol{x} \in \mathcal{X}$ such that $\langle \boldsymbol{y}, G(\boldsymbol{x}) \rangle \geq 0$. The upshot is that EAH enables us to solve (2) with just the above tools. Indeed, consider the following problem, which is an $\epsilon$-approximate version of the dual of (2).

$$\text{find} \quad \boldsymbol{y} \in \mathcal{Y} \quad \text{s.t.} \quad \langle \boldsymbol{y}, G(\boldsymbol{x}) \rangle \leq -\epsilon \quad \forall \boldsymbol{x} \in \mathcal{X}. \tag{3}$$

Since a GER oracle exists, (3) is infeasible. What is more, a certificate of infeasibility of (3) yields an $\epsilon$-approximate solution to (2). It thus suffices to run the ellipsoid algorithm on (3) and extract a certificate of infeasibility; in a nutshell, this is what EAH does (*cf.* Algorithm 1 in Appendix C).

**Theorem 2.1** (Generalized form of EAH; Farina & Pipis, 2024). *Given a* GER *oracle and a separation oracle (*SEP*)*

*for $\mathcal{Y}$,* EAH *runs in time* $\operatorname{poly}(d, m, \log(1/\epsilon))$ *and returns an $\epsilon$-approximate solution to* (2).

One of our main results (Theorem 4.1) crucially hinges on a strengthening of Theorem 2.1 due to Daskalakis et al. (2025), discussed further in Section 4 and Appendix C.3.

## 3. Existence and Complexity Barriers

Perhaps the most basic question about $\Phi$-EVIs concerns their *totality*—the existence of solutions. If one is willing to tolerate an arbitrarily small imprecision $\epsilon > 0$, we show that solutions exist under very broad conditions.

**Theorem 3.1.** *Suppose that $F : \mathcal{X} \to \mathbb{R}^d$ is measurable and there exists $L > 0$ such that every $\phi \in \Phi$ is $L$-Lipschitz continuous. Then, for any $\epsilon > 0$, there exists an $\epsilon$-approximate solution to the $\Phi$-EVI problem.*

In particular, our existence proof does not rest on $F$ being continuous. Instead, we consider the continuous function $\widehat{F}$ that maps $\boldsymbol{x} \mapsto \mathbb{E}_{\widehat{\boldsymbol{x}} \sim \Delta(\mathcal{B}_\delta(\boldsymbol{x}) \cap \mathcal{X})} F(\widehat{\boldsymbol{x}})$ (Claim C.1), where $\mathcal{B}_\delta(\boldsymbol{x})$ is the Euclidean ball centered at $\boldsymbol{x}$ with radius $\delta = \delta(\epsilon)$. It then suffices to invoke Brouwer's fixed-point theorem for the gradient mapping $\boldsymbol{x} \mapsto \Pi_{\mathcal{X}}(\boldsymbol{x} - \widehat{F}(\boldsymbol{x}))$, where $\Pi_{\mathcal{X}}$ is the Euclidean projection with respect to $\mathcal{X}$.

Theorem 3.1 implies that a $\Phi$-EVI can have approximate solutions even when the associated VI problem does not.[2]

**Corollary 3.2.** *There exists a VI problem that does not admit approximate solutions when $\epsilon = \Theta(1)$, but the corresponding $\epsilon$-approximate $\Phi$-EVI is total for any $\epsilon > 0$.*

In the proof, we set $F$ to be the *sign function* (Example C.2). By contrast, if one insists on exact solutions, EVIs do not necessarily admit solutions.

**Proposition 3.3.** *When $F$ is not continuous, there exists an EVI problem with no solutions.*

Furthermore, Theorem 3.1 raises the question of whether it is enough to instead assume that every $\phi \in \Phi$ is continuous. Our next result dispels any such hopes.

**Theorem 3.4.** *There are $\Phi$-EVI instances that do not admit $\epsilon$-approximate solutions even when $\epsilon = \Theta(1)$, $F$ is piecewise constant, and $\Phi$ contains only continuous functions.*

Our final result on existence complements Theorems 3.1 and 3.4 by showing that when $\Phi$ is finite-dimensional, it is enough if every $\phi \in \Phi$ admits a fixed point (this holds, for example, when $\phi$ is continuous—by Brouwer's theorem).

**Theorem 3.5.** *Suppose that*

1. *$\Phi$ is finite-dimensional, that is, there exists $k \in \mathbb{N}$ and a kernel map $m : \mathcal{X} \to \mathbb{R}^k$ such that every $\phi \in \Phi$ can be expressed as $\mathbf{K}m(\boldsymbol{x})$ for some $\mathbf{K} \in \mathbb{R}^{d \times k}$; and*

2. *every $\phi \in \Phi$ admits a fixed point, that is, a point $\mathcal{X} \ni \boldsymbol{x} = \mathrm{FP}(\phi)$ such that $\phi(\boldsymbol{x}) = \boldsymbol{x}$.*

*Then, the $\Phi$-EVI problem admits an $\epsilon$-approximate solution with support size at most $1 + dk$ for every $\epsilon > 0$.*

Notably, this theorem guarantees the existence of solutions with finite support; the proof makes use of the minimax theorem (*e.g.*, Sion, 1958) in conjunction with Carathéodory's theorem on convex hulls (Carathéodory, 1911).

**Complexity** Having established some basic existence properties, we now turn to the complexity of $\Phi$-EVIs. Let us define the VI gap function $\mathrm{VIGap}(\boldsymbol{x}) := -\min_{\boldsymbol{x}' \in \mathcal{X}} \langle F(\boldsymbol{x}), \boldsymbol{x}' - \boldsymbol{x} \rangle$, which is nonnegative. If we place no restrictions on $\Phi$, it turns out that $\Phi$-EVIs are tantamount to regular VIs:

**Proposition 3.6.** *If $\Phi$ contains all measurable functions from $\mathcal{X}$ to $\mathcal{X}$, then any solution $\mu \in \Delta(\mathcal{X})$ to the $\epsilon$-approximate $\Phi$-EVI problem satisfies*

$$\mathbb{E}_{\boldsymbol{x} \sim \mu} \mathrm{VIGap}(\boldsymbol{x}) \leq \epsilon. \tag{4}$$

In proof, it suffices to consider a $\phi$ that maps $\boldsymbol{x} \in \mathcal{X}$ to an appropriate point in $\operatorname{argmin}_{\boldsymbol{x}' \in \mathcal{X}} \langle F(\boldsymbol{x}), \boldsymbol{x}' - \boldsymbol{x} \rangle$. When $\mu$ must be given explicitly, Proposition 3.6 immediately implies that $\Phi$-EVIs are computationally hard, because (4) implies that $\mathrm{VIGap}(\boldsymbol{x}) \leq \epsilon$ for some $\boldsymbol{x}$ in the support of $\mu$, and such a point can be identified in polynomial time.[3]

**Corollary 3.7.** *The $\epsilon$-approximate $\Phi$-EVI problem is* PPAD-*hard even when $\epsilon$ is an absolute constant and $F$ is linear.*

Coupled with Proposition 3.6, this follows from the hardness result of Rubinstein (2015) concerning Nash equilibria in (multi-player) polymatrix games (for binary-action, graphical games, Deligkas et al., 2023 recently showed that PPAD-hardness persists up to $\epsilon < 1/2$). Corollary 3.7 notwithstanding, it is easy to see that the set of solutions to $\Phi$-EVIs is convex for any $\Phi \subseteq \mathcal{X}^{\mathcal{X}}$.

*Remark* 3.8. Let $\mathcal{X} = \mathcal{X}_1 \times \cdots \times \mathcal{X}_n$, as in an $n$-player game. Whether Corollary 3.7 applies under deviations that can be decomposed as $\phi : \boldsymbol{x} \mapsto \phi(\boldsymbol{x}) = (\phi_1(\boldsymbol{x}_1), \ldots, \phi_n(\boldsymbol{x}_n))$ is a major open question in the regime where $\epsilon \ll 1$ (*cf.* Dagan et al., 2024; Peng & Rubinstein, 2024).

---

[2]Noncontinuity of $F$ manifests itself prominently in *nonsmooth* optimization (*e.g.*, Zhang et al., 2020; Davis et al., 2022; Tian et al., 2022; Jordan et al., 2023a); recent research there focuses on *Goldstein* stationary points (Goldstein, 1977), which are conceptually related to EVIs.

[3]This argument carries over without restricting the support of $\mu$, by assuming instead access to a sampling oracle from $\mu$: a standard Chernoff bound implies that the empirical distribution (w.r.t. a large enough sample size) approximately satisfies (4).

Viewed differently, a special case of the $\Phi$-EVI problem arises when $\Phi = \{\phi\}$ and $F(\boldsymbol{x}) = \boldsymbol{x} - \phi(\boldsymbol{x})$, for some fixed map $\phi : \mathcal{X} \to \mathcal{X}$. In this case, the $\Phi$-EVI problem reduces to finding a $\mu \in \Delta(\mathcal{X})$ such that

$$\mathbb{E}_{\boldsymbol{x} \sim \mu} \langle F(\boldsymbol{x}), \phi(\boldsymbol{x}) - \boldsymbol{x} \rangle = - \mathbb{E}_{\boldsymbol{x} \sim \mu} \|\phi(\boldsymbol{x}) - \boldsymbol{x}\|^2 \geq -\epsilon. \quad (5)$$

As a result, $\mu$ must contain in its support an $\epsilon$-approximate fixed point of $\phi$, a problem which is PPAD-hard already for quadratic functions (Zhang et al., 2024a).

**Corollary 3.9.** *The $\epsilon$-approximate $\Phi$-EVI problem is* PPAD-*hard even when $\epsilon$ is an absolute constant, $F$ is quadratic, and $\Phi = \{\phi\}$ for a quadratic map $\phi : \mathcal{X} \to \mathcal{X}$.*

It is also worth noting that, unlike Corollary 3.7, $\Phi$ in the corollary above contains only continuous functions.

It also follows from (5) that for $\epsilon = 0$, $\Phi$-EVIs capture exact fixed points. The complexity class FIXP characterizes such problems (Etessami & Yannakakis, 2007).

**Corollary 3.10.** *The $\Phi$-EVI problem is* FIXP-*hard, assuming that* $\mathrm{supp}(\mu) \leq \mathrm{poly}(d)$.

Exponential lower bounds in terms of the number of function evaluations of $F$ also follow from Hirsch et al. (1989).

On a positive note, the next section establishes polynomial-time algorithms when $\Phi$ contains only *linear* endomorphisms.[4]

# 4. Efficient Computation with Linear Maps

The hardness results of the previous section highlight the need to restrict the set $\Phi$ in order to make meaningful progress. Our main result here establishes a polynomial-time algorithm when $\Phi$ contains only linear endomorphisms.

**Theorem 4.1.** *If $\Phi$ contains only linear endomorphisms, the $\epsilon$-approximate $\Phi$-EVI problem can be solved in time* $\mathrm{poly}(d, \log(B/\epsilon))$ *given a membership oracle for $\mathcal{X}$.*

The proof relies on the ellipsoid against hope (EAH), and in particular, a recent generalization by Daskalakis et al. (2025). In a nutshell, the main deficiency in the framework covered earlier in Section 2.1 is that one needs a separation oracle for $\mathcal{Y}$ (Theorem 2.1), where $\mathcal{Y}$ for us is the set of deviations $\Phi$. Unlike some applications, in which $\mathcal{Y}$ has an explicit, polynomial representation (Papadimitriou & Roughgarden, 2008), that assumption needs to be relaxed to account for $\Phi_{\mathsf{LIN}}$ (Daskalakis et al., 2025, Theorem 3.4).

Daskalakis et al. (2025) address this by considering instead the SEPorGER oracle. As the name suggests, for any

---

[4]We do not distinguish between affine and linear maps because we can always set $\mathcal{X} \leftarrow \mathcal{X} \times \{1\}$, in which case affine and linear maps coincide.

$\boldsymbol{y} \in \mathbb{R}^m$, it *either* returns a hyperplane separating $\boldsymbol{y}$ from $\mathcal{Y}$, or a good-enough-response $\boldsymbol{x} \in \mathcal{X}$. They showed that Theorem 2.1 can be extended under this weaker oracle (in place of GER and SEP); the formal version is given in Theorem C.4.

In our setting, we consider the feasibility problem

$$\text{find} \quad \phi \in \Phi_{\mathsf{LIN}} \quad \text{s.t.} \quad (6)$$
$$\langle F(\boldsymbol{x}), \phi(\boldsymbol{x}) - \boldsymbol{x} \rangle \leq -\epsilon \quad \forall \boldsymbol{x} \in \mathcal{X}.$$

Equivalently,

$$\text{find} \quad \mathbf{K} \in \mathbb{R}^{d \times d} \quad \text{s.t.}$$
$$\langle F(\boldsymbol{x}), \mathbf{K}\boldsymbol{x} - \boldsymbol{x} \rangle \leq -\epsilon \quad \forall \boldsymbol{x} \in \mathcal{X},$$
$$\mathbf{K}\boldsymbol{x} \in \mathcal{X} \quad \forall \boldsymbol{x} \in \mathcal{X}.$$

This program is infeasible since, for any $\phi \in \Phi_{\mathsf{LIN}}$, the fixed point $\boldsymbol{x}$ of $\phi$ makes the left-hand side of the constraint 0. And a certificate of infeasibility is an $\epsilon$-approximate $\Phi_{\mathsf{LIN}}$-EVI solution. Thus, it suffices to show how to run the ellipsoid algorithm on (6). By Theorem C.4, it suffices if for any $\mathbf{K} \in \mathbb{R}^{d \times d}$, we can compute efficiently *either*

- some $\boldsymbol{x} \in \mathcal{X}$ such that $\mathbf{K}\boldsymbol{x} = \boldsymbol{x}$ (GER), *or*
- some hyperplane separating $\mathbf{K}$ from $\Phi_{\mathsf{LIN}}$ (SEP).

This is precisely the *semi-separation oracle* solved by Daskalakis et al. (2025, Lemma 4.1), stated below.

**Lemma 4.2** (Daskalakis et al., 2025). *There is an algorithm that takes $\mathbf{K} \in \mathbb{R}^{d \times d}$, runs in $\mathrm{poly}(d)$ time, makes $\mathrm{poly}(d)$ oracle queries to $\mathcal{X}$, and either returns a fixed point $\mathcal{X} \ni \boldsymbol{x} = \mathbf{K}\boldsymbol{x}$, or a hyperplane separating $\mathbf{K}$ from $\Phi_{\mathsf{LIN}}$.*

On a separate note, Theorem 4.1 only accounts for approximate solutions. We cannot hope to improve that in the sense that exact solutions might be supported only on irrational points even in concave maximization (*cf.* Proposition 6.3).

## 4.1. Regret Minimization for EVIs on Polytopes

One caveat of Theorem 4.1 is that it relies on the impractical EAH algorithm. To address this limitation, we will show that $\Phi$-EVIs are also amenable to the more scalable approach of *regret minimization*—albeit with an inferior complexity growing as $\mathrm{poly}(1/\epsilon)$.

Specifically, in our context, the regret minimization framework can be applied as follows. At any time $t \in \mathbb{N}$, we think of a "learner" selecting a point $\boldsymbol{x}^{(t)} \in \mathcal{X}$, whereupon $F(\boldsymbol{x}^{(t)})$ is given as feedback from the "environment," so that the utility at time $t$ reads $-\langle \boldsymbol{x}^{(t)}, F(\boldsymbol{x}^{(t)}) \rangle$. $\Phi$-*regret* is a measure of performance in online learning, defined as

$$\Phi\text{-Reg}^{(T)} := \max_{\phi \in \Phi} \sum_{t=1}^{T} \langle F(\boldsymbol{x}^{(t)}), \phi(\boldsymbol{x}^{(t)}) - \boldsymbol{x}^{(t)} \rangle.$$

The uniform distribution $\mu$ on $\{\boldsymbol{x}^{(1)}, \dots, \boldsymbol{x}^{(T)}\}$ is clearly a $\Phi\text{-Reg}^{(T)}/T$-approximate $\Phi$-EVI solution.

In what follows, we will assume that $\mathcal{X}$ is a polytope given explicitly by linear constraints, *i.e.*,

$$\mathcal{X} = \{\boldsymbol{x} \in \mathbb{R}^d : \mathbf{A}\boldsymbol{x} \leq \boldsymbol{b}\},$$

where $\mathbf{A} \in \mathbb{Q}^{m \times d}$ and $\boldsymbol{b} \in \mathbb{Q}^m$ are given as input.

To minimize $\Phi$-regret, we will make use of the template by Gordon et al. (2008), which comprises two components. The first is a fixed-point oracle, which takes as input a function $\phi \in \Phi_{\mathsf{LIN}}$ and returns a point $\boldsymbol{x} \in \mathcal{X}$ with $\boldsymbol{x} = \phi(\boldsymbol{x})$; given that $\phi$ is linear, it can be implemented efficiently via linear programming. The second component is an algorithm for minimizing (external) regret over the set $\Phi_{\mathsf{LIN}}$. In Theorem D.1, we devise a polynomial representation for $\Phi_{\mathsf{LIN}}$:

**Theorem 4.3.** *For an arbitrary polytope $\mathcal{X}$ given by explicit linear constraints, there is an explicit representation of $\Phi_{\mathsf{LIN}}$ as a polytope with $O(d^2 + m^2)$ variables and constraints.*

As a consequence, we can instantiate the regret minimizer operating over $\Phi_{\mathsf{LIN}}$ with projected gradient descent.

**Corollary 4.4.** *There is a deterministic algorithm that guarantees $\Phi_{\mathsf{LIN}}\text{-}\mathsf{Reg}^{(T)} \leq \epsilon$ after $\mathrm{poly}(d, m)/\epsilon^2$ rounds, and requires solving a convex quadratic program with $O(d^2 + m^2)$ variables and constraints in each iteration.*

An additional benefit of Corollary 4.4 compared to using EAH is that the former is more suitable in a decentralized environment—for example, in multi-player games (*cf.* Example 5.1). There, Corollary 4.4 corresponds to each player running their own independent no-regret learning algorithm. Even in this setting, our algorithms actually yield an improvement over the best-known algorithms for minimizing $\Phi_{\mathsf{LIN}}$-regret over explicitly-represented polytopes: the previous state of the art, due to Daskalakis et al. (2025), requires running the ellipsoid algorithm on each iteration, which is slower than quadratic programming (Appendix D).

## 5. Game Theory Applications of EVIs

A major motivation for studying $\Phi$-EVIs lies in a strong connection to *(C)CEs* (Aumann, 1974; Moulin & Vial, 1978) in games. Indeed, we begin this section by pointing out that $\Phi$-EVIs capture (C)CEs for specific choices of $\Phi$.

We will mostly consider $n$-player *concave* games. Here, each player $i \in [n]$ selects a strategy $\boldsymbol{x}_i \in \mathcal{X}_i$ from some convex and compact set $\mathcal{X}_i$, and its utility is given by $u_i : (\boldsymbol{x}_1, \ldots, \boldsymbol{x}_n) \mapsto \mathbb{R}$. We assume that $u_i(\boldsymbol{x}_i, \boldsymbol{x}_{-i})$ is differentiable and concave in $\boldsymbol{x}_i$ for any $\boldsymbol{x}_{-i}$, and that the gradients $\nabla_{\boldsymbol{x}_i} u_i(\boldsymbol{x}_i, \boldsymbol{x}_{-i})$ are bounded. We let $\mathcal{X} := \mathcal{X}_1 \times \cdots \times \mathcal{X}_n$.

**Example 5.1** (CCE)**.** A distribution $\mu \in \Delta(\mathcal{X})$, is an $\epsilon$-*coarse correlated equilibrium (CCE)* (Moulin & Vial, 1978) if for any player $i \in [n]$,

$$\delta_i := \max_{\boldsymbol{x}_i' \in \mathcal{X}_i} \mathop{\mathbb{E}}_{\boldsymbol{x} \sim \mu} u_i(\boldsymbol{x}_i', \boldsymbol{x}_{-i}) - \mathop{\mathbb{E}}_{\boldsymbol{x} \sim \mu} u_i(\boldsymbol{x}) \leq \epsilon. \quad (7)$$

Now, consider an $\epsilon$-approximate EVI solution $\mu$ of the problem defined by

$$F := (-\nabla_{\boldsymbol{x}_1} u_1(\boldsymbol{x}), \ldots, -\nabla_{\boldsymbol{x}_n} u_n(\boldsymbol{x})).$$

Such $\mu$ satisfies, by concavity, $\sum_{i=1}^n \delta_i \leq \epsilon$; it is not necessarily an $\epsilon$-approximate CCE since it is possible that for some $i \in [n]$, *all* deviations strictly decrease $i$'s utility (so that $\delta_i$ in (7) is negative)—$\mu$ is technically an *average* CCE in the parlance of Nadav & Roughgarden (2010). To capture CCE via $\Phi$-EVIs, one can instead consider a richer set of deviations of the form $(\boldsymbol{x}_1, \ldots, \boldsymbol{x}_n) \mapsto (\boldsymbol{x}_1, \ldots, \boldsymbol{x}_i', \ldots, \boldsymbol{x}_n)$ for all $i \in [n]$ and $\boldsymbol{x}_i' \in \mathcal{X}_i$.

A canonical example of the above formalism is a *normal-form game*, in which each constraint set $\mathcal{X}_i$ is the probability simplex $\Delta(\mathcal{A}_i)$ over a finite set of *actions* $\mathcal{A}_i$, and each utility $u_i$ is a multilinear function.

**Example 5.2** (LCE)**.** A distribution $\mu \in \Delta(\mathcal{X})$ is an $\epsilon$-*linear correlated equilibrium (LCE)* if for any $i \in [n]$,

$$\max_{\phi_i \in \Phi_i} \mathop{\mathbb{E}}_{\boldsymbol{x} \sim \mu} u_i(\phi_i(\boldsymbol{x}_i), \boldsymbol{x}_{-i}) - \mathop{\mathbb{E}}_{\boldsymbol{x} \sim \mu} u_i(\boldsymbol{x}) \leq \epsilon,$$

where $\Phi_i$ contains all linear functions from $\mathcal{X}_i$ to $\mathcal{X}_i$. To capture LCE via $\Phi$-EVIs, it suffices to consider deviations of the form $(\boldsymbol{x}_1, \ldots, \boldsymbol{x}_n) \mapsto (\boldsymbol{x}_1, \ldots, \phi_i(\boldsymbol{x}_i), \ldots, \boldsymbol{x}_n)$ for all $i \in [n]$ and $\phi_i \in \Phi_i$.

For normal-form games, LCEs amount to the usual notion of CEs (Aumann, 1974). LCEs were introduced in the context of extensive-form games (Farina & Pipis, 2023; 2024).

**Refining correlated equilibria** In fact, and more surprisingly, $\Phi_{\mathsf{LIN}}$-EVI solutions can be a strict subset of LCEs.[5] This separation can already be appreciated in the setting of normal-form games, and manifests itself in at least two distinct ways. First, there exist games for which a CE need not be a solution to the $\Phi_{\mathsf{LIN}}$-EVI. In this sense, $\Phi_{\mathsf{LIN}}$-EVIs yield a computationally tractable superset of Nash equilibria that is tighter than CEs. Second, computation suggests that the set of solutions of the $\Phi_{\mathsf{LIN}}$-EVI for the game need not be a polyhedron, unlike the set of CEs. We provide a graphical depiction of this phenomenon in Figure 1. The figure depicts the set of $\Phi_{\mathsf{LIN}}$-EVI solutions to a simple "Bach or Stravinsky" game, in which the players receive payoffs $(3, 2)$ if they both pick Bach, $(2, 3)$ if they both pick Stravinsky, and $(0, 0)$ otherwise.

---

[5]The example of Şeref Ahunbay (2025, Example 1) already implies that certain CEs can be excluded from the set of $\Phi_{\mathsf{LIN}}$-EVIs, which, incidentally, could have implications for last-iterate convergence in some classes of games, as discussed by that author. Our example in Figure 1 goes much further, revealing that $\Phi_{\mathsf{LIN}}$-EVIs can yield significantly different utilities for each player compared to CEs.

**Interpretation** The reason for this separation is that, for a map $\phi : \mathcal{X} \to \mathcal{X}$, each player's mapped strategy $\phi(\boldsymbol{x})_i$ can also depend (linearly) on *other players' strategies* $\boldsymbol{x}_{-i}$. Indeed, the EVI formulation of a game does not take into account the identities of the players. For this reason, we will call the set of $\Phi_{\mathsf{LIN}}$-EVI solutions in a concave game *anonymous linear correlated equilibria*, or *ALCE* for short. We give two game-theoretic interpretations of ALCEs.

First, the ALCEs of a game $\Gamma$ are the *symmetric* LCEs of the "symmetrized" game in which the players are randomly shuffled before the game begins. That is, consider the $n$-player game $\Gamma^{\mathrm{sym}}$ defined as follows. Each player's strategy set is $\mathcal{X}$. For strategy profile $(\boldsymbol{x}^1, \dots, \boldsymbol{x}^n) \in \mathcal{X}^n$, the utility to player $i$ is given by

$$u_i^{\mathrm{sym}}(\boldsymbol{x}^1, \dots, \boldsymbol{x}^n) = \frac{1}{n!} \sum_{\sigma \in \mathfrak{G}_n} u_{\sigma(i)}(\boldsymbol{x}_1^{\sigma^{-1}(1)}, \dots, \boldsymbol{x}_n^{\sigma^{-1}(n)}),$$

where $\mathfrak{G}_n$ is the set of permutations $\sigma : [n] \to [n]$. The following result then follows almost by definition.

**Proposition 5.3.** *For a given distribution $\mu \in \Delta(\mathcal{X})$, define the distribution $\mu^n \in \Delta(\mathcal{X}^n)$ by sampling $\boldsymbol{x} \sim \mu$ and outputting $(\boldsymbol{x}, \dots, \boldsymbol{x}) \in \mathcal{X}^n$. Then, $\mu$ is a ALCE of $\Gamma$ if and only if $\mu^n$ is an LCE of $\Gamma^{\mathrm{sym}}$.*

Second, for normal-form games, the ALCEs are the distributions $\mu \in \Delta(\mathcal{X})$ such that no player $i$ has a profitable deviation of the following form. The correlation device first samples $\boldsymbol{x} \sim \mu$, and samples recommendations $a_j \sim \boldsymbol{x}_j$ for each player $j$. Then, the player selects another player $j$ (possibly $j = i$) whose recommendation it wishes to see. The player then observes a sample $a_j' \sim \boldsymbol{x}_j$ that is *independently* sampled from $a_j$.[6] Finally, the player chooses an action $a_i^* \in \mathcal{A}_i$, and each player $j$ gets reward $u_j(a_i^*, a_{-i})$. Thus, players are allowed (modulo the independent sampling) to *spy* on each others' recommendations.

Further discussion about ALCEs and formal proofs of the claims in this section are deferred to Appendix F.

**Coupled constraints** Continuing from Examples 5.1 and 5.2, we observe that $(\Phi_{\mathsf{LIN}}\text{-})$EVIs can be used even in "pseudo-games," in which $\mathcal{X}$ does not necessarily decompose into $\mathcal{X}_1 \times \cdots \times \mathcal{X}_n$; this means that $\boldsymbol{x}_i \in \mathcal{X}_i(\boldsymbol{x}_{-i})$. As we discuss in Appendix A, most prior work in such settings has focused on generalized Nash equilibria, with the exception of Bernasconi et al. (2023). $(\Phi_{\mathsf{LIN}}\text{-})$EVIs induce an interesting notion of LCE/CCE in pseudo-games, albeit not directly comparable to the one put forward by Bernasconi et al. (2023). It is worth noting that Bernasconi et al. (2023) left open whether efficient algorithms for computing their notion of (coarse) correlated equilibria exist.

---

[6]This independence is crucial: without it, $\mu$ would actually need to be a distribution over pure Nash equilibria!

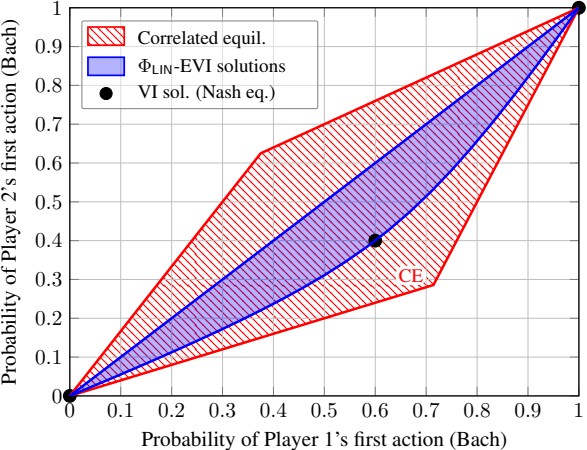

*Figure 1.* Marginals of the set of correlated equilibria (CE) and of the set of solutions to $\Phi_{\mathsf{LIN}}$-EVI in the simple $2 \times 2$ game "Bach or Stravinsky." The x- and y-axes show the probability with which the two players select the first action (Bach). The set of marginals of $\Phi_{\mathsf{LIN}}$-EVI solutions appears to have a curved boundary corresponding, we believe, to the hyperbola $10x^2 - 25xy + 10y^2 - 6x + 11y = 0$.

**Definition 5.4.** Given an $n$-player pseudo-game with concave, differentiable utilities and joint constraints $\mathcal{X}$, a distribution $\mu \in \Delta(\mathcal{X})$ is an $\epsilon$-*ALCE* if

$$\max_{\phi \in \Phi_{\mathsf{LIN}}} \mathbb{E}_{\boldsymbol{x} \sim \mu} \sum_{i=1}^n u_i(\phi(\boldsymbol{x})_i, \boldsymbol{x}_{-i}) - \sum_{i=1}^n u_i(\boldsymbol{x}) \le \epsilon.$$

By virtue of our main result (Theorem 4.1), such an equilibrium can be computed in polynomial time.

**Noncontinuous gradients** In fact, our results do not rest on the usual assumption that each player's gradient is a continuous function, thereby significantly expanding the scope of prior known results even in games. For example, we refer to Dasgupta & Maskin (1986); Bichler et al. (2021); Martin & Sandholm (2024) for pointers to some applications.

**Nonconcave games** Last but not least, $\Phi$-EVIs give rise to a notion of *local* $\Phi$-equilibrium (Definition G.1) in nonconcave games. It turns out that this captures recent results by Cai et al. (2024a) and Şeref Ahunbay (2025), but our framework has certain important advantages. First, we give a $\mathrm{poly}(d, \log(1/\epsilon))$-time algorithm (Theorem 4.1), while theirs scale polynomially in $1/\epsilon$. Second, our results do not assume continuity of the gradients. And finally, our algorithms are polynomial even when $\Phi$ contains all linear endomorphisms (Theorem 4.1). Appendix G elaborates further on those points.

# 6. Problems Where EVIs Coincide with VIs

We saw earlier, in Proposition 3.6, that when $\Phi$ comprises all functions from $\mathcal{X}$ to $\mathcal{X}$, the $\Phi$-EVI problem is tantamount to the associated VI problem. However, if one restricts the functions contained in $\Phi$, are there still structured VIs where we retain this equivalence? In this section, we consider certain structured VIs, and show their equivalence to the corresponding EVIs (that is, $\Phi_{\mathsf{CON}}$-EVIs). Unlike general VIs, the ones we examine below are tractable.

## 6.1. Polymatrix Zero-Sum Games and Beyond

The first important class of VIs we consider is described by a condition given below.

**Proposition 6.1.** *Suppose that for any $\boldsymbol{x}' \in \mathcal{X}$, the function $g : \boldsymbol{x} \mapsto \langle F(\boldsymbol{x}), \boldsymbol{x}' - \boldsymbol{x} \rangle$ is concave. Then, if $\mu \in \Delta(\mathcal{X})$ is an $\epsilon$-approximate solution to the EVI, $\mathbb{E}_{\boldsymbol{x} \sim \mu} \boldsymbol{x}$ is an $\epsilon$-approximate solution to the VI.*

The proof follows directly from Jensen's inequality.

The precondition of Proposition 6.1 is satisfied, *e.g.*, when: (i) $\langle F(\boldsymbol{x}), \boldsymbol{x} \rangle = 0$ for all $\boldsymbol{x} \in \mathcal{X}$, and (ii) $F$ is a linear map. In the context of $n$-player games, the first condition amounts to the zero-sum property: $\sum_{i=1}^{n} u_i(\boldsymbol{x}) = 0$ for all $\boldsymbol{x}$. Of course, this property is not enough to enable efficient computation of Nash equilibria, for every two-player (general-sum) game can be converted into a 3-player zero-sum game. This is where the second condition comes into play: $F$ is a linear map—that is, each player's gradient must be linear in the joint strategy. Those two conditions are satisfied in *polymatrix zero-sum* games (Cai et al., 2016); in such games, the conclusion of Proposition 6.1 is a well-known fact.

## 6.2. Quasar-Concave Functions

We next consider the problem of maximizing a (single) function that satisfies *quasar-concavity*—a natural generalization of concavity that has received significant interest (Hardt et al., 2018; Fu et al., 2023; Hinder et al., 2020; Gower et al., 2021; Guminov et al., 2023; Caramanis et al., 2024).[7]

**Definition 6.2** (Quasar-concavity). Let $\gamma \in (0, 1]$ and $\boldsymbol{x}^\star \in \mathcal{X}$ be a maximizer of a differentiable function $u : \mathcal{X} \to \mathbb{R}$. We say that $u$ is $\gamma$-*quasar-concave* with respect to $\boldsymbol{x}^\star$ if

$$u(\boldsymbol{x}^\star) \leq u(\boldsymbol{x}) + \frac{1}{\gamma} \langle \nabla u(\boldsymbol{x}), \boldsymbol{x}^\star - \boldsymbol{x} \rangle \quad \forall \boldsymbol{x} \in \mathcal{X}. \quad (8)$$

In particular, in the special case where $\gamma = 1$, (8) is equivalent to *star-concavity* (Nesterov & Polyak, 2006). If in addition (8) holds for all $\boldsymbol{x}^\star \in \mathcal{X}$ (not merely w.r.t. a global maximizer), it captures the usual notion of concavity.

---

[7]Prior literature mostly uses the term *quasar-convexity*, which is equivalent to quasar-concavity for the opposite function $-u$.

Any reasonable solution concept for such problems should place all mass on global maxima; EVIs pass this litmus test:

**Proposition 6.3.** *Let $F = -\nabla u$ for a $\gamma$-quasar-concave and differentiable function $u : \mathcal{X} \to \mathbb{R}$. Then, for any solution $\mu \in \Delta(\mathcal{X})$ to the EVI problem,*

$$\mathbb{E}_{\boldsymbol{x} \sim \mu} u(\boldsymbol{x}) \geq \max_{\boldsymbol{x} \in \mathcal{X}} u(\boldsymbol{x}).$$

*Thus, $\mathbb{P}_{\boldsymbol{x} \sim \mu}[u(\boldsymbol{x}^\star) = u(\boldsymbol{x})] = 1$, for $\boldsymbol{x}^\star \in \operatorname{argmax}_{\boldsymbol{x}} u(\boldsymbol{x})$.*

Indeed, by Definition 6.2, $0 \leq \mathbb{E}_{\boldsymbol{x} \sim \mu} \langle \nabla u(\boldsymbol{x}), \boldsymbol{x} - \boldsymbol{x}^\star \rangle \leq \gamma \mathbb{E}_{\boldsymbol{x} \sim \mu} [u(\boldsymbol{x}) - u(\boldsymbol{x}^\star)]$ for any EVI solution $\mu \in \Delta(\mathcal{X})$. Thus, under quasar-concavity, VIs basically reduce to EVIs.

# 7. Performance Guarantees for EVIs

In many settings, a VI solution is used as a proxy to approximately maximize some underlying objective function; machine learning offers many such applications. The question is whether performance guarantees pertaining to VIs can be extended—potentially with some small degradation—to EVIs as well. The purpose of this section is to provide a framework for achieving that based on the following notion.

**Definition 7.1.** An EVI problem is $(\lambda, \nu)$-*smooth*, for $\lambda > 0, \nu > -1$, w.r.t. $W : \mathcal{X} \to \mathbb{R}$ and $\boldsymbol{x}^\star \in \operatorname{argmax}_{\boldsymbol{x}} W(\boldsymbol{x})$ if

$$\langle F(\boldsymbol{x}), \boldsymbol{x}^\star - \boldsymbol{x} \rangle \leq -\lambda W(\boldsymbol{x}^\star) + (\nu + 1) W(\boldsymbol{x}) \quad \forall \boldsymbol{x} \in \mathcal{X}.$$

**Example 7.2.** When the underlying problem corresponds to a multi-player game and $W$ is the (utilitarian) social welfare, Definition 7.1 coincides with the celebrated notion of smoothness *à la* Roughgarden (2015); this is a consequence of multilinearity, which implies that $W(\boldsymbol{x}) = -\langle \boldsymbol{x}, F(\boldsymbol{x}) \rangle$ and $\langle \boldsymbol{x}^\star, F(\boldsymbol{x}) \rangle = -\sum_{i=1}^{n} u_i(\boldsymbol{x}_i^\star, \boldsymbol{x}_{-i})$ for all $\boldsymbol{x} \in \mathcal{X}$. We also refer to the recent treatment of smoothness by Şeref Ahunbay (2025) in the context of nonconcave games, which builds on the primal-dual framework of Nadav & Roughgarden (2010).

Definition 7.1 is an extension of the more general notion of "local smoothness," introduced by Roughgarden & Schoppmann (2015) in the context of splittable congestion games. However, it goes beyond games. Indeed, the following definition we introduce generalizes Definition 6.2, making a new connection between smoothness and quasar-concavity.

**Definition 7.3** (Extension of quasar-concavity). Let $\boldsymbol{x}^\star \in \mathcal{X}$ be a maximizer of a differentiable function $u : \mathcal{X} \to \mathbb{R}$. We say that $u$ is $(\lambda, \nu)$-*smooth* with respect to $\boldsymbol{x}^\star$ if

$$\langle \nabla u(\boldsymbol{x}), \boldsymbol{x}^\star - \boldsymbol{x} \rangle \geq \lambda u(\boldsymbol{x}^\star) - (\nu + 1) u(\boldsymbol{x}) \quad \forall \boldsymbol{x} \in \mathcal{X}.$$

In particular, when $\lambda := \gamma$ and $\nu := \gamma - 1$, the above definition captures $\gamma$-quasar-concavity. In Appendix E, we

provide an example of a polynomial that satisfies Definition 7.3 without being quasi-concave. Now, the key property of Definition 7.1 is that any EVI solution approximates the underlying objective—by a factor of $\rho := \lambda/1+\nu$.

**Theorem 7.4.** *Let $\mu \in \Delta(\mathcal{X})$ be an $\epsilon$-approximate solution to a $(\lambda, \nu)$-smooth EVI problem w.r.t. $W : \mathcal{X} \to \mathbb{R}$. Then,*

$$\mathbb{E}_{\boldsymbol{x} \sim \mu} W(\boldsymbol{x}) \geq \frac{\lambda}{1 + \nu} \max_{\boldsymbol{x} \in \mathcal{X}} W(\boldsymbol{x}) - \frac{\epsilon}{1 + \nu}.$$

The proof follows directly from Definition 7.1, using that $\mathbb{E}_{\boldsymbol{x} \sim \mu} \langle F(\boldsymbol{x}), \boldsymbol{x}^\star - \boldsymbol{x} \rangle \geq -\epsilon$ and linearity of expectation.

## 8. Conclusions and Future Research

In summary, our main contribution was to introduce and examine a natural relaxation of VIs, which we refer to as *expected* VIs. Unlike VIs, which are marred by computational intractability, we showed that EVIs can be solved efficiently under minimal assumptions. We also uncovered many other intriguing properties of EVIs (*cf.* Table 1).

There are many promising avenues for future work. VIs enjoy a great reach in a wide range of applications, some of which were discussed earlier in our introduction. It would be interesting to explore in more detail how EVIs fare in such settings compared to VIs. In particular, given that EVIs relax VIs, in addition to their more favorable computational properties, it is likely that they unlock new, more desirable solutions not present under VIs. For example, it is well known (*e.g.*, Ashlagi et al., 2005) that CEs can achieve better welfare than Nash equilibria in games. In light of the prominence of correlated equilibria in the rich setting of multi-player games, we anticipate EVIs to solidify their place also in other application areas beyond the realm of game theory.

## Impact Statement

This paper presents work whose goal is to advance the field of machine learning. There are many potential societal consequences of our work, none which we feel must be specifically highlighted here.

## Acknowledgments

T.S. is supported by the Vannevar Bush Faculty Fellowship ONR N00014-23-1-2876, National Science Foundation grants RI-2312342 and RI-1901403, ARO award W911NF2210266, and NIH award A240108S001. B.H.Z. is supported by the CMU Computer Science Department Hans Berliner PhD Student Fellowship. E.T, R.E.B., and V.C. thank the Cooperative AI Foundation, Polaris Ventures (formerly the Center for Emerging Risk Research) and Jaan Tallinn's donor-advised fund at Founders Pledge for financial support. E.T. and R.E.B. are also supported in part by the Cooperative AI PhD Fellowship. G.F is supported by the National Science Foundation grant CCF-2443068. We are grateful to Mete Şeref Ahunbay for his helpful feedback. We also thank Andrea Celli and Martino Bernasconi for discussions regarding Φ-equilibria in games with coupled constraints.

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

## A. Further Related Work

We have seen that Definition 1.2 is strongly connected with the notion of $\Phi$-equilibria. In extensive-form games, the question of characterizing the set of deviations $\Phi$ that enables efficient learning—within the no-regret framework—and computation has attracted considerable attention. In particular, efficient algorithms have been established for *extensive-form correlated equilibria (EFCEs)* (Huang & von Stengel, 2008; Farina et al., 2022; Morrill et al., 2021a;b), and more broadly, when $\Phi$ contains solely *linear* functions (Farina & Pipis, 2024; 2023)—corresponding to *linear correlated equilibria (LCEs)*. Recently, Daskalakis et al. (2025) strengthened those results beyond extensive-form games whenever there is a separation oracle for the constraint set; we rely on their approach for some of our positive results. Moreover, Zhang et al. (2024a; 2025) established certain positive results even when $\Phi$ contains low-degree polynomials; by contrast, in our setting, $\Phi$-EVIs are hard even when $\Phi$ contains only quadratic polynomials (Corollary 3.9).

Besides encompassing correlated equilibria in games, wherein the constraint set $\mathcal{X}$ can be decomposed as a Cartesian product over the constraint set of each player (reflecting the fact that players select strategies independently), our positive results pertaining to Definition 1.2 do not rest on such assumptions and apply even in the presence of joint constraint sets. There is a long history in game theory, optimization, and economics pertaining to such settings—sometimes referred to as "pseudo-games" in the literature (Goktas & Greenwald, 2022; Arrow & Debreu, 1954; Facchinei & Kanzow, 2010; Fischer et al., 2014; Facchinei et al., 2009; Ardagna et al., 2012; Tatarenko & Kamgarpour, 2018; Jordan et al., 2023b; Daskalakis et al., 2021). The notion of *generalized* Nash equilibria—the natural counterpart of Nash's solution concept in the presence of coupled constraints—has dominated that line of work, with the recent paper of Bernasconi et al. (2023) being the notable exception.

Responding to the call of Daskalakis (2022), Cai et al. (2024a) and Şeref Ahunbay (2025) recently proposed several tractable solution concepts in games with nonconcave utilities. In particular, when specialized to games, $\Phi$-EVIs are closely related to a notion proposed by Şeref Ahunbay (2025, Definition 6). One of our key results, Theorem 4.1, establishes a $\text{poly}(d, \log(1/\epsilon))$-time algorithm, while the algorithms of Cai et al. (2024b) and Şeref Ahunbay (2025) scale polynomially in $1/\epsilon$. Also, Theorem 4.1 applies even when $\Phi$ contains all linear endomorphisms (Theorem 4.1). From a more conceptual vantage point, a significant part of our contribution is to extend the scope of such results beyond games, as we have already highlighted.

Kapron & Samieefar (2024) and Bernasconi et al. (2024) recently studied the computational complexity of VIs, and generalizations thereof—namely, *quasi VIs*, establishing PPAD-completeness under mild assumptions. Whether our framework can be extended to encompass quasi VIs is left as an interesting direction for the future.

Finally, it would be remiss not to point out that our VI relaxation is in the spirit of "lifting," a standard technique whereby the original problem is *lifted* to a higher-dimensional space to gain more analytical and computational leverage; a concrete example, in the context of optimal transport theory, is Kantorovich's relaxation of Monge's formulation (Villani, 2009). Such techniques have been fruitful in the context of min-max optimization (Hsieh et al., 2019; Domingo-Enrich et al., 2020).

## B. Additional Preliminaries

**Revisiting Definition 1.2** In order to define the distributions $\Delta(\mathcal{X})$ over $\mathcal{X}$ precisely, we recall here some basic concepts from probability theory. We refer to Billingsley (1999, Chapter 1 and 2) and Aliprantis & Border (2006, Chapter 15) for detailed treatments. We assume throughout the paper that the set $\mathcal{X} \subseteq \mathbb{R}^d$ is Borel measurable. Let $\Delta(\mathcal{X})$ be the set of Borel probability measures $\mu$ on $\mathcal{X}$, that is, measures $\mu : (\mathcal{X}, B(\mathcal{X})) \to (\mathbb{R}, B(\mathbb{R}))$ with $\mu(\mathcal{X}) = 1$, where $B(\mathcal{X})$ and $B(\mathbb{R})$ denote the respective $\sigma$-algebra of Borel sets. We simply call $\mu$ a distribution. For any Borel measurable function $f : \mathcal{X} \to \mathbb{R}$—henceforth just *measurable*— we can then take the integral $\mathbb{E}_{\boldsymbol{x} \sim \mu}[f(\boldsymbol{x})] := \int_{\mathcal{X}} f(\boldsymbol{x}) d\mu(\boldsymbol{x})$. In particular, for $\mathbb{E}_{\boldsymbol{x} \sim \mu} \langle F(\boldsymbol{x}), \phi(\boldsymbol{x}) - \boldsymbol{x} \rangle$ in Definition 1.2 to be well-defined, we assume throughout this paper that $F$ and each $\phi \in \Phi$ are measurable functions.

For our computational results (Section 4), we are making a standard assumption regarding the geometry of $\mathcal{X}$ (Section 2); this can be met by bringing $\mathcal{X}$ into isotropic position. In particular, there is a polynomial-time algorithm that computes an affine transformation to accomplish that (Lovász & Vempala, 2006), and minimizing linear-swap regret reduces to minimizing linear-swap regret to the transformed instance (Daskalakis et al., 2025, Lemma A.1).

## C. Omitted Proofs

This section contains the proofs omitted from the main body.

### C.1. Existence of $\Phi$-EVI solutions

We begin with Theorem 3.1.

**Theorem 3.1.** *Suppose that $F : \mathcal{X} \to \mathbb{R}^d$ is measurable and there exists $L > 0$ such that every $\phi \in \Phi$ is $L$-Lipschitz continuous. Then, for any $\epsilon > 0$, there exists an $\epsilon$-approximate solution to the $\Phi$-EVI problem.*

*Proof.* We define a function $\widehat{F}_\delta : \mathcal{X} \to \mathbb{R}^d$ as

$$\widehat{F}_\delta : \boldsymbol{x} \mapsto \frac{1}{|\mathcal{B}_\delta(\boldsymbol{x}) \cap \mathcal{X}|} \int_{\mathcal{B}_\delta(\boldsymbol{x}) \cap \mathcal{X}} F(\widehat{\boldsymbol{x}}) d\nu(\widehat{\boldsymbol{x}});$$

this is a rescaled Lebesgue integral, which represents a multivariate local average. Above,

- $\delta > 0$ is a sufficiently small parameter, to be defined shortly;
- $\mathcal{B}_\delta(\boldsymbol{x}) \subseteq \mathbb{R}^d$ is the (closed) Euclidean ball of radius $\delta$ centered at $\boldsymbol{x}$; and
- $|\cdot|$ denotes the set's Borel measure.

Given that $F$ is assumed to be bounded, we can define $B \in \mathbb{R}$ such that $\max_{\boldsymbol{x} \in \mathcal{X}} \|F(\boldsymbol{x})\| \leq B$. For the proof below, it will suffice to set $\delta := \epsilon/(L+1)B$.

We first observe that $\widehat{F}_\delta$ is continuous.

**Claim C.1.** *$\widehat{F}_\delta$ is continuous.*

*Proof.* We will show that for any $\boldsymbol{x} \in \mathcal{X}$ and $\epsilon' > 0$, we can choose $\delta' = \delta'(\epsilon')$ such that for any $\boldsymbol{x}' \in \mathcal{X}$ with $\|\boldsymbol{x} - \boldsymbol{x}'\| < \delta'$,

$$\|\widehat{F}_\delta(\boldsymbol{x}) - \widehat{F}_\delta(\boldsymbol{x}')\| \leq \epsilon'.$$

By the triangle inequality, the difference $\|\widehat{F}_\delta(\boldsymbol{x}) - \widehat{F}_\delta(\boldsymbol{x}')\|$ can be decomposed as the sum of

$$Ⓐ := \left| \frac{1}{|\mathcal{B}_\delta(\boldsymbol{x}) \cap \mathcal{X}|} - \frac{1}{|\mathcal{B}_\delta(\boldsymbol{x}') \cap \mathcal{X}|} \right| \int_{\mathcal{B}_\delta(\boldsymbol{x}) \cap \mathcal{X}} \|F(\widehat{\boldsymbol{x}})\| d\nu(\widehat{\boldsymbol{x}})$$

and

$$Ⓑ := \frac{1}{|\mathcal{B}_\delta(\boldsymbol{x}') \cap \mathcal{X}|} \left\| \int_{\mathcal{B}_\delta(\boldsymbol{x}) \cap \mathcal{X}} F(\widehat{\boldsymbol{x}}) d\nu(\widehat{\boldsymbol{x}}) - \int_{\mathcal{B}_\delta(\boldsymbol{x}') \cap \mathcal{X}} F(\widehat{\boldsymbol{x}}) d\nu(\widehat{\boldsymbol{x}}) \right\|.$$

Now, Ⓐ can be bounded as

$$Ⓐ \leq B \left| 1 - \frac{|\mathcal{B}_\delta(\boldsymbol{x}) \cap \mathcal{X}|}{|\mathcal{B}_\delta(\boldsymbol{x}') \cap \mathcal{X}|} \right| \leq \frac{1}{2}\epsilon',$$

where we selected $\delta'$ small enough so that

$$\left(1 - \frac{\epsilon'}{B}\right) |\mathcal{B}_\delta(\boldsymbol{x}') \cap \mathcal{X}| \leq |\mathcal{B}_\delta(\boldsymbol{x}) \cap \mathcal{X}| \leq \left(1 + \frac{\epsilon'}{B}\right) |\mathcal{B}_\delta(\boldsymbol{x}') \cap \mathcal{X}|.$$

Moreover, by selecting $\delta'$ small enough so that

$$|(\mathcal{B}_\delta(\boldsymbol{x}) \cap \mathcal{X}) \setminus (\mathcal{B}_\delta(\boldsymbol{x}') \cap \mathcal{X})| + |(\mathcal{B}_\delta(\boldsymbol{x}') \cap \mathcal{X}) \setminus (\mathcal{B}_\delta(\boldsymbol{x}) \cap \mathcal{X})| \leq \frac{1}{2B}\epsilon' |\mathcal{B}_\delta(\boldsymbol{x}') \cap \mathcal{X}|, \tag{9}$$

we have

$$\left\| \int_{\mathcal{B}_\delta(\boldsymbol{x}) \cap \mathcal{X}} F(\widehat{\boldsymbol{x}}) d\nu(\widehat{\boldsymbol{x}}) - \int_{\mathcal{B}_\delta(\boldsymbol{x}') \cap \mathcal{X}} F(\widehat{\boldsymbol{x}}) d\nu(\widehat{\boldsymbol{x}}) \right\|$$

$$\leq \int_{(\mathcal{B}_\delta(\boldsymbol{x}) \cap \mathcal{X}) \setminus (\mathcal{B}_\delta(\boldsymbol{x}') \cap \mathcal{X})} \|F(\widehat{\boldsymbol{x}})\| d\nu(\widehat{\boldsymbol{x}}) + \int_{(\mathcal{B}_\delta(\boldsymbol{x}') \cap \mathcal{X}) \setminus (\mathcal{B}_\delta(\boldsymbol{x}) \cap \mathcal{X})} \|F(\widehat{\boldsymbol{x}})\| d\nu(\widehat{\boldsymbol{x}})$$

$$\leq B|(\mathcal{B}_\delta(\boldsymbol{x}) \cap \mathcal{X}) \setminus (\mathcal{B}_\delta(\boldsymbol{x}') \cap \mathcal{X})| + B|(\mathcal{B}_\delta(\boldsymbol{x}') \cap \mathcal{X}) \setminus (\mathcal{B}_\delta(\boldsymbol{x}) \cap \mathcal{X})| \leq \frac{1}{2}\epsilon' |\mathcal{B}_\delta(\boldsymbol{x}') \cap \mathcal{X}|,$$

where the last inequality uses (9). As a result, we have shown that Ⓐ+Ⓑ $\leq \epsilon'$, thereby implying that $\|\widehat{F}_\delta(\boldsymbol{x}) - \widehat{F}_\delta(\boldsymbol{x}')\| \leq \epsilon'$. This completes the proof. □

Having established that $\widehat{F}_\delta$ is continuous, we can now apply Brouwer's fixed point theorem on the map $\boldsymbol{x} \mapsto \Pi_{\mathcal{X}}(\boldsymbol{x} - \widehat{F}_\delta(\boldsymbol{x}))$, where we recall that $\Pi_{\mathcal{X}}$ denotes the Euclidean projection onto $\mathcal{X}$. This implies that there is a point $\boldsymbol{x} \in \mathcal{X}$ such that $\boldsymbol{x} = \Pi_{\mathcal{X}}(\boldsymbol{x} - \widehat{F}_\delta(\boldsymbol{x}))$. Moreover, such a point satisfies the VI constraint with respect to $\widehat{F}_\delta$:

$$\langle \widehat{F}_\delta(\boldsymbol{x}), \boldsymbol{x}' - \boldsymbol{x} \rangle \geq 0 \quad \boldsymbol{x}' \in \mathcal{X};$$

for example, see Kinderlehrer & Stampacchia (2000, Section 3) for the derivation. Finally, we define $\mu \in \Delta(\mathcal{X})$ to be the uniform distribution over $\mathcal{B}_\delta(\boldsymbol{x}) \cap \mathcal{X}$. Then, for any $\phi \in \Phi$,

$$\langle \widehat{F}_\delta(\boldsymbol{x}), \phi(\boldsymbol{x}) - \boldsymbol{x} \rangle = \underset{\widehat{\boldsymbol{x}} \sim \mu}{\mathbb{E}} \langle F(\widehat{\boldsymbol{x}}), \phi(\boldsymbol{x}) - \boldsymbol{x} \rangle$$
$$= \underset{\widehat{\boldsymbol{x}} \sim \mu}{\mathbb{E}} \langle F(\widehat{\boldsymbol{x}}), \widehat{\boldsymbol{x}} - \boldsymbol{x} \rangle + \underset{\widehat{\boldsymbol{x}} \sim \mu}{\mathbb{E}} \langle F(\widehat{\boldsymbol{x}}), \phi(\boldsymbol{x}) - \phi(\widehat{\boldsymbol{x}}) \rangle + \underset{\widehat{\boldsymbol{x}} \sim \mu}{\mathbb{E}} \langle F(\widehat{\boldsymbol{x}}), \phi(\widehat{\boldsymbol{x}}) - \widehat{\boldsymbol{x}} \rangle. \tag{10}$$

The first term in (10) can be bounded as

$$\underset{\widehat{\boldsymbol{x}} \sim \mu}{\mathbb{E}} \langle F(\widehat{\boldsymbol{x}}), \widehat{\boldsymbol{x}} - \boldsymbol{x} \rangle \leq \sqrt{\underset{\widehat{\boldsymbol{x}} \sim \mu}{\mathbb{E}} \|F(\widehat{\boldsymbol{x}})\|^2} \sqrt{\underset{\widehat{\boldsymbol{x}} \sim \mu}{\mathbb{E}} \|\widehat{\boldsymbol{x}} - \boldsymbol{x}\|^2} \leq \delta B, \tag{11}$$

where we used the Cauchy-Schwarz inequality, the fact that $\|F(\widehat{\boldsymbol{x}})\| \leq B$ for all $\widehat{\boldsymbol{x}} \in \mathcal{X}$, and $\|\widehat{\boldsymbol{x}} - \boldsymbol{x}\| \leq \delta$ for all $\widehat{\boldsymbol{x}}$ in the support of $\mu$. Similarly, the second term in (10) can be bounded as

$$\underset{\widehat{\boldsymbol{x}} \sim \mu}{\mathbb{E}} \langle F(\widehat{\boldsymbol{x}}), \phi(\boldsymbol{x}) - \phi(\widehat{\boldsymbol{x}}) \rangle \leq \sqrt{\underset{\widehat{\boldsymbol{x}} \sim \mu}{\mathbb{E}} \|F(\widehat{\boldsymbol{x}})\|^2} \sqrt{\underset{\widehat{\boldsymbol{x}} \sim \mu}{\mathbb{E}} \|\phi(\widehat{\boldsymbol{x}}) - \phi(\boldsymbol{x})\|^2}$$
$$\leq L \sqrt{\underset{\widehat{\boldsymbol{x}} \sim \mu}{\mathbb{E}} \|F(\widehat{\boldsymbol{x}})\|^2} \sqrt{\underset{\widehat{\boldsymbol{x}} \sim \mu}{\mathbb{E}} \|\widehat{\boldsymbol{x}} - \boldsymbol{x}\|^2} \leq \delta B L, \tag{12}$$

where we additionally used the assumption that $\phi$ is $L$-Lipschitz continuous. Combining (11) and (12) with (10), we have

$$\underset{\widehat{\boldsymbol{x}} \sim \mu}{\mathbb{E}} \langle F(\widehat{\boldsymbol{x}}), \phi(\widehat{\boldsymbol{x}}) - \widehat{\boldsymbol{x}} \rangle \geq -\delta(L+1)B + \langle \widehat{F}_\delta(\boldsymbol{x}), \phi(\boldsymbol{x}) - \boldsymbol{x} \rangle \geq -\delta(L+1)B,$$

and this holds for any $\phi \in \Phi$. Setting $\delta := \epsilon/(L+1)B$ completes the proof. □

We next proceed with Corollary 3.2 and Proposition 3.3, which are restated below.

**Corollary 3.2.** *There exists a VI problem that does not admit approximate solutions when $\epsilon = \Theta(1)$, but the corresponding $\epsilon$-approximate $\Phi$-EVI is total for any $\epsilon > 0$.*

**Proposition 3.3.** *When $F$ is not continuous, there exists an EVI problem with no solutions.*

We provide an example that will establish both of those claims.

**Example C.2** (Discontinuous $F$; *cf.* Corollary 3.2 and Proposition 3.3). Let $F(x)$ be the sign function,

$$F(x) = \mathrm{sgn}(x) := \begin{cases} -1 & \text{if} \quad x < 0, \\ 1 & \text{otherwise}, \end{cases}$$

and $\mathcal{X} = [-1, 1]$. We first claim that there is no $\epsilon$-approximate VI solution for $\epsilon < 1$. Indeed,

- for any $x < 0$, picking $x' = 1$ ensures $\langle F(x), x' - x \rangle = x - 1 < -1$;
- for any $x \geq 0$, picking $x' = -1$ ensures $\langle F(x), x' - x \rangle = -1 - x \leq -1$.

There is also no *exact* EVI solution to this problem. Indeed, consider any $\mu \in \Delta(\mathcal{X})$.

- If $\mathbb{P}_{x \sim \mu}[x = 0] = 1$, then taking $x' = -1$ ensures $\mathbb{E}_{x \sim \mu}\langle F(x), x' - x \rangle = \langle F(0), x' \rangle = -1$.
- Otherwise, taking $x' = 0$, we have

$$\underset{x \sim \mu}{\mathbb{E}} \langle F(x), x' - x \rangle = \underset{x \sim \mu}{\mathbb{E}} [-|x|] < 0.$$

On the other hand, for any $\epsilon > 0$, there exists an $\epsilon$-approximate EVI solution (as promised by Theorem 3.1). In particular, suppose that $\mu$ uniformly picks between $-\epsilon$ and $\epsilon$. Then, for any $x' \in \mathcal{X}$,

$$\mathbb{E}_{x \sim \mu} \langle F(x), x' - x \rangle = -\frac{1}{2}(x' + \epsilon) + \frac{1}{2}(x' - \epsilon) = -\epsilon.$$

It is worth pointing out that the above example can be slightly modified so that exact EVI solutions do exist, as we explain below.

**Example C.3** (Modification of Example C.2 with exact VI)**.** We define $F(x)$ identically to Example C.2, except $F(1/2) = -1$. We claim that there is no VI solution for $\epsilon < 1/2$: any $x \neq 1/2$ by Example C.2, and $x = 1/2$ is not a solution since $y = 1$ ensures $\langle F(x), x' - x \rangle = -1/2$.

However, there is an exact EVI solution: fix any $x^\star \in [0, 1/2)$ and consider $\mu$ that uniformly mixes between $x = x^\star$ and $x = 1/2$. Then, for any $x' \in \mathcal{X}$,

$$\mathbb{E}_{x \sim \mu} \langle F(x), x' - x \rangle = \frac{1}{2}(x' - x^\star) - \frac{1}{2}\left(x' - \frac{1}{2}\right) = \frac{1}{2}\left(\frac{1}{2} - x^\star\right) > 0.$$

Our next result reveals that the precondition of Theorem 3.1 with respect to $\Phi$ cannot be relaxed to continuity.

**Theorem 3.4.** *There are $\Phi$-EVI instances that do not admit $\epsilon$-approximate solutions even when $\epsilon = \Theta(1)$, $F$ is piecewise constant, and $\Phi$ contains only continuous functions.*

*Proof.* As before, let $F(x)$ be the sign function,

$$F(x) = \mathrm{sgn}(x) := \begin{cases} -1 & \text{if } x < 0, \\ 1 & \text{otherwise,} \end{cases}$$

and $\mu \in \Delta([-1, 1])$ be any distribution. For $\delta > 0$, let $\phi_\delta : [-1, 1] \to [-1, 1]$ be given by

$$\phi_\delta(x) = \begin{cases} 1 & \text{if } x < -2\delta, \\ -(x + \delta)/\delta & \text{if } -2\delta \leq x \leq 0, \\ -1 & \text{if } x > 0. \end{cases}$$

Further, let $\phi_0(x) := -\mathrm{sgn}(x)$. Every $\phi_\delta$ (with $\delta > 0$) is continuous, by construction. Now, note that $\phi_\delta \to \phi_0$ pointwise when $\delta \downarrow 0$, and every $\phi_\delta$ is bounded. As a result, by the dominated convergence theorem, we have

$$\lim_{\delta \to 0} \mathbb{E}_{x \sim \mu}[F(x)(\phi_\delta(x) - x)] = \mathbb{E}_{x \sim \mu}[F(x)(\phi_0(x) - x)]$$
$$= \mathbb{E}_{x \sim \mu}[-1 - F(x) \cdot x] \leq -1,$$

where the last line uses the fact that $F(x)\phi_0(x) = -\mathrm{sgn}(x)^2 = -1$ and $F(x) \cdot x = \mathrm{sgn}(x) \cdot x = |x|$ for all $x$. Thus, for any $\epsilon < 1$, there must be some $\delta > 0$ for which $\mathbb{E}[F(x)(\phi_\delta(x) - x)] < -\epsilon$, so $\mu$ cannot be an $\epsilon$-approximate EVI solution. $\square$

Continuing on Section 3, we next provide the proof of Theorem 3.5.

**Theorem 3.5.** *Suppose that*

1. *$\Phi$ is finite-dimensional, that is, there exists $k \in \mathbb{N}$ and a kernel map $m : \mathcal{X} \to \mathbb{R}^k$ such that every $\phi \in \Phi$ can be expressed as $\mathbf{K}m(\boldsymbol{x})$ for some $\mathbf{K} \in \mathbb{R}^{d \times k}$; and*

2. *every $\phi \in \Phi$ admits a fixed point, that is, a point $\mathcal{X} \ni \boldsymbol{x} = \mathrm{FP}(\phi)$ such that $\phi(\boldsymbol{x}) = \boldsymbol{x}$.*

*Then, the $\Phi$-EVI problem admits an $\epsilon$-approximate solution with support size at most $1 + dk$ for every $\epsilon > 0$.*

*Proof.* We assume, without loss of generality, that (as functions) the coordinates $m_i : \mathcal{X} \to \mathbb{R}$ for $1 \leq i \leq k$ are linearly independent. We further assume that $m$ is bounded, again without loss of generality. (Indeed, if for example $m_i$ is unbounded, then column $i$ of $\mathbf{K}$ must contain all zeros, or else $\phi_{\mathbf{K}}(\boldsymbol{x}) := \mathbf{K}m(\boldsymbol{x})$ would be unbounded; we can thus freely remove such coordinates $m_i$.)

Now, let $\mathcal{K} := \operatorname{conv}\{\mathbf{K} : \phi_{\mathbf{K}} \in \Phi\}$ be the set of matrices corresponding to maps in $\Phi$; we can assume that $\mathcal{K}$ is closed. We can now rewrite the $\Phi$-EVI problem as

$$\text{find} \quad \mu \in \Delta(\mathcal{X}) \quad \text{s.t.} \quad \mathop{\mathbb{E}}_{\boldsymbol{x} \sim \mu} \left\langle F(\boldsymbol{x})m(\boldsymbol{x})^\top, \mathbf{K} - \mathbf{I} \right\rangle \geq 0$$

for all $\mathbf{K} \in \mathcal{K}$, where above $\mathbf{I}$ is the identity matrix and the inner product is the usual Frobenius inner product of matrices.[8] Further, let $\mathcal{A} := \operatorname{conv}\{F(\boldsymbol{x})m(\boldsymbol{x})^\top : \boldsymbol{x} \in \mathcal{X}\}$. Then, the $\Phi$-EVI problem can be in turn expressed as

$$\text{find} \quad \mathbf{A} \in \mathcal{A} \quad \text{s.t.} \quad \langle \mathbf{A}, \mathbf{K} - \mathbf{I} \rangle \geq 0$$

for all $\mathbf{K} \in \mathcal{K}$. Since $F$ and $m$ are bounded, by assumption, so is $\mathcal{A}$. Moreover, since the coordinates $m_i$ are linearly independent, $\mathcal{K}$ is also bounded. Thus, letting $\bar{\mathcal{A}}$ denote the closure of $\mathcal{A}$, the max-min problem

$$\max_{\mathbf{A} \in \bar{\mathcal{A}}} \min_{\mathbf{K} \in \mathcal{K}} \langle \mathbf{A}, \mathbf{K} - \mathbf{I} \rangle \tag{13}$$

satisfies the conditions of the minimax theorem. Moreover, for any $\mathbf{K} \in \mathcal{K}$, the fixed point $\boldsymbol{x} := \operatorname{FP}(\phi_{\mathbf{K}})$ satisfies

$$\left\langle F(\boldsymbol{x})m(\boldsymbol{x})^\top, \mathbf{K} - \mathbf{I} \right\rangle = \langle F(\boldsymbol{x}), \phi_{\mathbf{K}}(\boldsymbol{x}) - \boldsymbol{x} \rangle = 0,$$

so the zero-sum game (13) has a nonnegative value; that is, there exists $\mathbf{A} \in \bar{\mathcal{A}}$ such that $\min_{\mathbf{K} \in \mathcal{K}}\langle \mathbf{A}, \mathbf{K} - \mathbf{I} \rangle \geq 0$. Thus, for every $\epsilon > 0$, there exists $\mathbf{A} \in \mathcal{A}$ such that $\min_{\mathbf{K} \in \mathcal{K}}\langle \mathbf{A}, \mathbf{K} - \mathbf{I} \rangle \geq -\epsilon$. Moreover, by Carathéodory's theorem, $\mathbf{A}$ can be expressed as a convex combination of at most $1 + dk$ matrices of the form $F(\boldsymbol{x})m(\boldsymbol{x})^\top$. This convex combination is thus an $\epsilon$-approximate EVI solution. $\square$

The only reason the above proof breaks when $\epsilon = 0$ is that $\mathcal{A}$ may not be closed. Indeed, this issue is fundamental: there are instances where no exact EVI solutions exist even when $\phi$ contains only constant functions (Proposition 3.3).

### C.2. Complexity of $\Phi$-EVIs

With regard to the complexity of computing $\Phi$-EVI solutions, the key observation is that, when $\Phi$ contains all (measurable) maps, $\Phi$-EVIs are essentially equivalent to VIs; this immediately implies a number of hardness results, which were covered earlier in Section 3. We provide the formal proof of Proposition 3.6 below.

**Proposition 3.6.** *If $\Phi$ contains all measurable functions from $\mathcal{X}$ to $\mathcal{X}$, then any solution $\mu \in \Delta(\mathcal{X})$ to the $\epsilon$-approximate $\Phi$-EVI problem satisfies*

$$\mathop{\mathbb{E}}_{\boldsymbol{x} \sim \mu} \operatorname{VIGap}(\boldsymbol{x}) \leq \epsilon. \tag{4}$$

*Proof.* We can define a measurable map $\phi : \mathcal{X} \to \mathcal{X}$ such that $\phi(\boldsymbol{x})$ is an element selected from $\operatorname{argmin}_{\boldsymbol{x}' \in \mathcal{X}}\langle F(\boldsymbol{x}), \boldsymbol{x}' - \boldsymbol{x} \rangle$ by utilizing the measurable maximum theorem (Aliprantis & Border, 2006, Theorem 18.19). To satisfy the conditions of this theorem, we need to define—using Aliprantis & Border's notation— the weakly measurable set-valued function $\psi : \mathcal{X} \rightrightarrows \mathcal{X}$ as $\psi(\boldsymbol{x}) = \mathcal{X}$ and the (Carathéodory) function $f : \mathcal{X} \times \mathcal{X} \to \mathbb{R}$ as $f(\boldsymbol{x}, \boldsymbol{x}') = -\langle F(\boldsymbol{x}), \boldsymbol{x}' - \boldsymbol{x} \rangle$. Due to this map $\phi$, a $\Phi$-EVI solution $\mu \in \Delta(\mathcal{X})$ must then, in particular, satisfy

$$\mathop{\mathbb{E}}_{\boldsymbol{x} \sim \mu} \langle F(\boldsymbol{x}), \phi(\boldsymbol{x}) - \boldsymbol{x} \rangle = \mathop{\mathbb{E}}_{\boldsymbol{x} \sim \mu} \operatorname*{argmin}_{\boldsymbol{x}' \in \mathcal{X}} \langle F(\boldsymbol{x}), \boldsymbol{x}' - \boldsymbol{x} \rangle \geq 0.$$

Therefore, there must exist $\boldsymbol{x}^\star \in \mathcal{X}$ with $\operatorname{argmin}_{\boldsymbol{x}' \in \mathcal{X}}\langle F(\boldsymbol{x}^\star), \boldsymbol{x}' - \boldsymbol{x}^\star \rangle \geq 0$, that is, a VI solution $\boldsymbol{x}^\star$. If $\mu$ has finite support, then such a $\boldsymbol{x}^\star$ exists within that support. The $\epsilon$-approximation case follows analogously. $\square$

---

[8]To avoid measurability issues, it is enough to consider here only distributions $\mu$ with finite support.

## C.3. Proof of Theorem 4.1

To establish Theorem 4.1, we will use the recent framework of Daskalakis et al. (2025), which refines Theorem 2.1 in the context of Section 2.1, ultimately summarized in Theorem C.4. Coupled with the "semi-separation oracle" of Lemma 4.2, we will thus arrive at Theorem 4.1.

Let $\mathcal{X} \subseteq \mathbb{R}^d$ and $\mathcal{Y} \subseteq \mathbb{R}^m$ be convex and compact sets. The goal is to solve the convex program

$$\text{find} \quad \mu \in \Delta(\mathcal{X}) \quad \text{s.t.} \quad \min_{\boldsymbol{y} \in \mathcal{Y}} \langle \mu, \mathbf{A}\boldsymbol{y} \rangle \geq 0, \tag{14}$$

where $\Delta(\mathcal{X}) \subseteq \mathbb{R}^M$ and $\mathbf{A} \in \mathbb{R}^{M \times m}$; we think of $M$ as being potentially exponentially large, so $\mathbf{A}$ is not given explicitly; $\Phi_{\mathsf{LIN}}$-EVIs can be expressed as (14), assuming that $\mu$ has finite support (*cf.* Theorem 3.5). The target is to solve (14) with complexity polynomial in $d$ and $m$ (and other parameters of the problem, except $M$). As we saw earlier in Section 2.1, the EAH algorithm accomplishes that given access to a GER oracle, which, for any $\boldsymbol{y} \in \mathcal{Y}$, returns $\boldsymbol{x} \in \mathcal{X}$ such that $\langle \mu(\boldsymbol{x}), \mathbf{A}\boldsymbol{y} \rangle \geq 0$, where $\Delta(\mathcal{X}) \ni \mu(\boldsymbol{x})$ places all probability on $\boldsymbol{x}$. Assuming that such an oracle exists, the convex program

$$\text{find} \quad \boldsymbol{y} \in \mathbb{R}_{>0}\mathcal{Y} \quad \text{s.t.} \quad \max_{\mu \in \Delta(\mathcal{X})} \langle \mu, \mathbf{A}\boldsymbol{y} \rangle \leq -1 \tag{15}$$

is infeasible, where $\mathbb{R}_{>0}\mathcal{Y} := \{c\boldsymbol{y} : \boldsymbol{y} \in \mathcal{Y}, c > 0\}$ is the conic hull of $\mathcal{Y}$. Despite its infeasibility, EAH proceeds by applying the ellipsoid algorithm on (15)—this is where the name "ellipsoid against hope" comes from. In doing so, the ellipsoid will eventually shrink to an area with negligible volume (denoted by vol), at which point one can extract a certificate of infeasibility for (15) as follows. The execution of the ellipsoid will have produced a sequence of $T \in \mathbb{N}$ good-enough-responses, $\boldsymbol{x}^{(1)}, \ldots, \boldsymbol{x}^{(T)}$, such that for any $\boldsymbol{y} \in \mathcal{Y}$, it holds that $\langle \mu(\boldsymbol{x}^{(t)}), \mathbf{A}\boldsymbol{y} \rangle \geq 0$ for some $t \in [T]$ (up to numerical imprecision). In turn, this implies that there is a mixture $\mu$ over $\{\boldsymbol{x}^{(1)}, \ldots, \boldsymbol{x}^{(T)}\}$ that guarantees $\langle \mu, \mathbf{A}\boldsymbol{y} \rangle \geq 0$ for every $\boldsymbol{y} \in \mathcal{Y}$. Such a $\mu$ can be computed in polynomial time by solving a smaller program, which simply searches over the mixing coefficients.

So far, we have elaborated on the framework presented in Section 2.1. To solve $\Phi_{\mathsf{LIN}}$-EVIs, it is necessary to relax the oracle assumed above. In particular, the SEPorGER oracle, introduced by Daskalakis et al. (2024), proceeds as follows. It takes as input a point $\boldsymbol{y} \in \mathbb{R}^m$ (not necessarily in $\mathcal{Y}$), and must *either* return a good-enough-response $\boldsymbol{x} \in \mathcal{X}$, or a hyperplane separating $\boldsymbol{y}$ from $\mathcal{Y}$. The idea now is to again run ellipsoid on (15), but by replacing $\mathcal{Y}$ with a convex "shell set"; every time the SEPorGER oracle returns a separating hyperplane, the shell set restricts further. At the end of this process, once the ellipsoid has shrank enough, one can work with the induced shell set $\widetilde{\mathcal{Y}}$ and proceed by identifying a mixture among the good-enough-responses $\{\boldsymbol{x}^{(1)}, \ldots, \boldsymbol{x}^{(T)}\}$ that approximately solves (14). The overall scheme is given in Algorithm 1.

Below, we state the main guarantee of Algorithm 1 shown by Daskalakis et al. (2025); in its statement, we have made certain slight adjustments in accordance with our setting.

**Theorem C.4** (Daskalakis et al., 2025). *Suppose that the following conditions hold.*

1. *$\mathbf{A} \in \mathbb{R}^{M \times m}$ such that for any $\mu \in \Delta(\mathcal{X})$, $\|\mu^\top \mathbf{A}\| \leq B$ for some $B \geq 1$;*

2. *$\mathcal{Y}$ is convex and compact, and satisfies $\mathcal{B}_{r_y}(\cdot) \subseteq \mathcal{Y} \subseteq \mathcal{B}_{R_y}(\mathbf{0})$; and*

3. *there exists a SEPorGER oracle: for every point $\boldsymbol{y} \in \mathcal{B}_{R_y}(\mathbf{0})$, it runs in $\mathrm{poly}(d, m)$ time, and either returns a hyperplane separating $\boldsymbol{y}$ from $\mathcal{Y}$ or a good-enough-response $\boldsymbol{x} \in \mathcal{X}$.*

*Then, Algorithm 1 runs in $\mathrm{poly}(d, m, \log(B/\epsilon))$ time and computes $\mu \in \Delta(\mathcal{X})$ such that*

$$\min_{\boldsymbol{y} \in \mathcal{Y}} \langle \mu, \mathbf{A}\boldsymbol{y} \rangle \geq -\epsilon.$$

That the second precondition (Item 2) is satisfied follows from Daskalakis et al. (2025, Lemma 2.3). The third precondition, Item 3, is satisfied by virtue of Lemma 4.2. Consequently, Theorem 4.1 follows from Theorem C.4.

*Remark* C.5 (Weak oracles and finite precision). Since we are working with general convex sets, the oracles posited in Section 2 (namely, membership, separation, and linear optimization) can return irrational outputs. This can be addressed by employing *weak* versions of those oracles, which relax the output by allowing some small slackness $\epsilon$ (Grötschel et al., 1993). Theorem 4.1 can be readily extended under those weaker oracles; see Daskalakis et al. (2025, Appendices E and F).

---

**Algorithm 1** Ellipsoid against hope (EAH) under `SEPorGER` oracle ([Daskalakis et al., 2025](#))

---

**input** • Parameters $R_y, r_y > 0$ such that $\mathcal{B}_{r_y}(\cdot) \subseteq \mathcal{Y} \subseteq \mathcal{B}_{R_y}(\mathbf{0})$

      • precision parameter $\epsilon > 0$

      • constant $B \geq 1$ such that $\|\mu^\top \mathbf{A}\| \leq B$ for all $\mu \in \Delta(\mathcal{X})$

      • a `SEPorGER` oracle

**output** A sparse, $\epsilon$-approximate solution $\mu \in \Delta(\mathcal{X})$ of (14)

1: Initialize the ellipsoid $\mathcal{E} := \mathcal{B}_{R_y}(\mathbf{0})$

2: Initialize $\widetilde{\mathcal{Y}} := \mathcal{B}_{R_y}(\mathbf{0})$

3: **while** $\mathrm{vol}(\mathcal{E}) \geq \mathrm{vol}(\mathcal{B}_{\epsilon/B}(\cdot))$ **do**

4:      Query the `SEPorGER` oracle on the center of $\mathcal{E}$

5:      **if** it returns a good-enough-response $\boldsymbol{x} \in \mathcal{X}$ **then**

6:          Update $\mathcal{E}$ to the minimum volume ellipsoid containing $\mathcal{E} \cap \{\boldsymbol{y} \in \mathbb{R}^m : \langle \boldsymbol{y}, \mathbf{A}^\top \mu(\boldsymbol{x}) \rangle \leq 0\}$

7:      **else**

8:          Let $H$ be the halfspace that separates $\boldsymbol{y}$ from $\mathcal{Y}$

9:          Update $\mathcal{E}$ to the minimum volume ellipsoid containing $\mathcal{E} \cap H$

10:        Update $\widetilde{\mathcal{Y}} := \widetilde{\mathcal{Y}} \cap H$

11:      **end if**

12: **end while**

13: Let $\boldsymbol{x}^{(1)}, \ldots, \boldsymbol{x}^{(T)}$ be the `GER` oracle responses produced in the process above

14: Define $\mathbf{X} := [\mu(\boldsymbol{x}^{(1)}) \mid \ldots \mid \mu(\boldsymbol{x}^{(T)})]$ and compute $\mathbf{X}^\top \mathbf{A} \in \mathbb{R}^{T \times m}$

15: Compute a solution $\boldsymbol{\lambda}$ to the convex program

$$\text{find} \quad \boldsymbol{\lambda} \in \Delta^T \quad \text{s.t.} \quad \min_{\boldsymbol{y} \in \widetilde{\mathcal{Y}}} \boldsymbol{\lambda}^\top (\mathbf{X}^\top \mathbf{A}) \boldsymbol{y} \geq -\epsilon$$

16: **return** $\Delta(\mathcal{X}) \ni \mu := \sum_{t=1}^T \lambda^{(t)} \mu(\boldsymbol{x}^{(t)})$

---

## D. Characterizing Linear Endomorphisms for Polytopes

In this section, we answer the following question. Given a nonempty polytope $\mathcal{X} = \{\boldsymbol{x} \in \mathbb{R}^d : \mathbf{A}\boldsymbol{x} \leq \boldsymbol{b}\}$ where $\mathbf{A} \in \mathbb{R}^{m \times d}$ and $\boldsymbol{b} \in \mathbb{R}^m$, we wish to characterize the set of (affine) linear maps $\phi : \mathcal{X} \to \mathcal{X}$. That is, we wish to understand the set of pairs $(\mathbf{K}, \boldsymbol{c}) \in \mathbb{R}^{d \times d} \times \mathbb{R}^d$ such that $\mathbf{K}\boldsymbol{x} + \boldsymbol{c} \in \mathcal{X}$ for all $\boldsymbol{x} \in \mathcal{X}$. The following result provides an explicit polynomial representation for that set, establishing [Theorem 4.3](#).

**Theorem D.1.** $\mathbf{K}\boldsymbol{x} + \boldsymbol{c} \in \mathcal{X}$ *for all* $\boldsymbol{x} \in \mathcal{X}$ *if and only if there is a matrix* $\mathbf{V} \in \mathbb{R}^{m \times m}$ *satisfying the constraints*

$$\mathbf{VA} = \mathbf{AK}, \quad \mathbf{V}\boldsymbol{b} \leq \boldsymbol{b} - \mathbf{A}\boldsymbol{c}, \quad \mathbf{V} \geq \mathbf{0}.$$

*Proof.* Let $\mathbf{K} \in \mathbb{R}^{d \times d}$ and $\boldsymbol{c} \in \mathbb{R}^d$, and let $\boldsymbol{a}_i^\top \boldsymbol{x} \leq b_i$ be the $i$th constraint that defines $\mathcal{X}$. Then, the claim that $\boldsymbol{a}_i^\top (\mathbf{K}\boldsymbol{x} + \boldsymbol{c}) \leq b_i$ for every $\boldsymbol{x} \in \mathcal{X}$ is equivalent to the claim that the linear program

$$\max_{\boldsymbol{x}} \quad \boldsymbol{a}_i^\top \mathbf{K}\boldsymbol{x} \quad \text{s.t.} \quad \mathbf{A}\boldsymbol{x} \leq \boldsymbol{b} \tag{16}$$

has value at most $b_i - \boldsymbol{a}_i^\top \boldsymbol{c}$. By strong duality, (16) has the same value as

$$\min_{\boldsymbol{v}_i} \quad \boldsymbol{b}^\top \boldsymbol{v}_i \quad \text{s.t.} \quad \mathbf{A}^\top \boldsymbol{v}_i = \mathbf{K}^\top \boldsymbol{a}_i, \quad \boldsymbol{v} \geq \mathbf{0}.$$

The theorem follows now by setting $\mathbf{V} = \begin{bmatrix} \boldsymbol{v}_1 & \ldots & \boldsymbol{v}_k \end{bmatrix}^\top$. $\qquad\square$

Furthermore, assuming that $\mathcal{B}_1(\mathbf{0}) \subseteq \mathcal{X} \subseteq \mathcal{B}_R(\mathbf{0})$ with $R \leq \mathrm{poly}(d)$, it follows that $\|\mathbf{K}\|_2, \|\mathbf{V}\|_2 \leq \mathrm{poly}(d)$, where $\|\cdot\|_2$ denotes the spectral norm. Indeed, to begin with, $\|\boldsymbol{c}\|_2 \leq R$ since $\mathbf{K} \cdot \mathbf{0} + \boldsymbol{c} \in \mathcal{X} \subseteq \mathcal{B}_r(\mathbf{0})$. For $\|\mathbf{K}\|_2$, take any $\boldsymbol{x} \in \mathbb{R}^d$ with $\|\boldsymbol{x}\| = 1$. Since $\mathcal{B}_1(\mathbf{0}) \subseteq \mathcal{X}$, we have $\boldsymbol{x} \in \mathcal{X}$, in turn implying that $\mathbf{K}\boldsymbol{x} + \boldsymbol{c} \in \mathcal{X}$. As a result, $\|\mathbf{K}\boldsymbol{x}\| - \|\boldsymbol{c}\| \leq \|\mathbf{K}\boldsymbol{x} + \boldsymbol{c}\| \leq \mathrm{poly}(d)$, from which it follows that $\|\mathbf{K}\|_2 \leq \mathrm{poly}(d)$. Further, one can take each $\boldsymbol{a}_i$ and $b_i$ to be such that $1 \leq b_i \leq \mathrm{poly}(d)$ and $\|\boldsymbol{a}_i\| = 1$, and so the bound $\|\mathbf{V}\|_2 \leq \mathrm{poly}(d)$ follows from the fact that $\mathbf{V}\boldsymbol{b} \leq \boldsymbol{b} - \mathbf{A}\boldsymbol{c}$ and $\mathbf{V} \geq \mathbf{0}$.

Combining these bounds with Theorem D.1, and as we saw earlier in Corollary 4.4, we are able to use standard techniques for minimizing regret over $\Phi_{\mathsf{LIN}}$—such as projected gradient descent.

For comparison, let us point out the approach of Daskalakis et al. (2025) for the case where $\mathcal{X}$ is given explicitly. To do so, we recall the following definition.

**Definition D.2.** We say that a polytope $\mathcal{X}$ has an *H-representation* of size $m$ if it is given as the intersection of $m$ halfspaces: $\mathcal{X} = \{x \in \mathbb{R}^d : \mathbf{A}x \leq b\}$ for some $\mathbf{A} \in \mathbb{Q}^{m \times d}$ and $b \in \mathbb{Q}^m$. It has a *V-representation* of size $m$ if it is given as the convex hull of $m$ vertices: $\mathcal{X} = \mathrm{conv}(\{v_1, \ldots, v_m\})$ for $v_1, \ldots, v_m \in \mathbb{Q}^d$.

In this context, they make the following crucial observation (Daskalakis et al., 2025, Lemmas 3.1 and 3.2).

**Lemma D.3** (Daskalakis et al., 2025)**.** *If $\mathcal{X}$ has either an $H$-representation of size $m$ or a $V$-representation of size $m$, there is a $\mathrm{poly}(d, m)$-time membership oracle for $\Phi_{\mathsf{LIN}}$.*

Using a membership oracle for $\Phi_{\mathsf{LIN}}$, it is also possible to construct a linear optimization oracle (Grötschel et al., 1993). As a result, coupled with Lemma D.3, standard algorithms—such as *follow-the-perturbed-leader* (Hazan, 2016)—can be applied to minimize regret over $\Phi_{\mathsf{LIN}}$. However, the main limitation is that constructing a linear optimization oracle using a membership oracle relies on the ellipsoid algorithm, which is impractical. In contrast, Theorem D.1 allows us to bypass using the ellipsoid algorithm, resulting in a more practical approach.

It is also worth noting that one can extend Theorem 4.3 using only a membership oracle for $\mathcal{X}$ (even when $\mathcal{X}$ is not an explicitly represented polytope) using techniques from Daskalakis et al. (2025), although the resulting algorithm is more elaborate and requires running the ellipsoid algorithm on every iteration to compute the next strategies.

# E. An Illustrative Example of Definition 7.3

In Section 7, we introduced a generalized notion of smoothness (Definition 7.1) that captures Roughgarden's notion in the context of multi-player games. As a result, there are numerous interesting examples that fall under Definition 7.1; for example, Roughgarden et al. (2017) provide a survey in the context of auctions. Our goal here is to provide a single function that satisfies Definition 7.3, but without being quasar-concave (in the sense of Definition 6.2).

**Example E.1.** We consider the polynomial function

$$u : x \mapsto -\frac{3}{4}px^4 + px^3 + 1, \tag{17}$$

where $p \in (0, 8]$. $u$ has a global maximum at $x = 1$, with value $1 + p/4$. It also admits a VI solution (in fact, a saddle point) at $x = 0$. This implies that $u$ is not $\gamma$-quasar-concave for any $\gamma \in (0, 1]$. On the other hand, it is not hard to verify the following claim.

**Claim E.2.** $u$ is $(1, p/4)$-smooth (Definition 7.3) for any $p \in (0, 8]$.

A graphical illustration of $u$ for various values of $p$ is given in Figure 2. Coupled with Theorem 7.4, Claim E.2 implies that any solution $\mu$ to the induced EVI problem satisfies

$$\mathbb{E}_{x \sim \mu} u(x) \geq \frac{1}{1 + \frac{p}{4}} \max u(x) = 1.$$

This guarantee is tight, since $x = 0$, with $u(0) = 1$, is a solution to the (E)VI problem.

*Remark* E.3. Definition 7.1, to which Definition 7.3 is a special case, is a generalization of smoothness in the sense of Roughgarden & Schoppmann (2015). While our bound applies to any EVI solution (Theorem 7.4), Roughgarden & Schoppmann (2015) gave a counter-example that excludes CCEs; this is not a contradiction because they define CCEs *without* linearizing the utilities (as in Example 5.1), while EVIs always operate over the linearized utilities.

# F. Omitted Details from Section 5

In this section and the next, we will use the notation $\Phi_{\mathsf{LIN}}(\mathcal{X}, \mathcal{Y})$ to denote the set of linear maps $\phi : \mathcal{X} \to \mathcal{Y}$.

In this section, let $\Gamma$ be a concave game. For each player $i$ let $\mathcal{X}_i \subset \mathbb{R}^{d_i}$ be its (convex, compact) strategy set, and let $\Phi_i \subseteq \mathcal{X}_i^{\mathcal{X}}$. We assume $u_i(\cdot, x_{-i})$ is differentiable in $x_i$ for all $i$.[9] Without loss of generality we assume that the projection

---

[9] By "$f : \mathcal{C} \to \mathbb{R}$ is differentiable" when $\mathcal{C}$ is closed, we mean that $f$ is defined and differentiable on an open set $\hat{\mathcal{C}} \supset \mathcal{C}$.

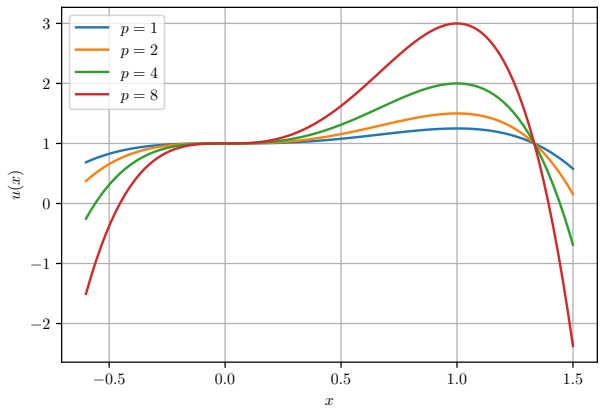

Figure 2. Function $u$, defined in (17), for $p \in \{1, 2, 4, 8\}$.

function $\pi_i(\boldsymbol{x}) = \boldsymbol{x}_i$ is in $\Phi_i$, and that $\Phi_i$ is convex. Crucially for this section and departing to our knowledge from all prior work on $\Phi$-equilibria in games, functions $\phi_i \in \Phi_i$ are allowed to depend not just on $\boldsymbol{x}_i$ but also on other $\boldsymbol{x}_{-i}$s. We first generalize the examples in Section 5 to arbitrary $\Phi$.

**Definition F.1.** An $\epsilon$-*approximate* $(\Phi_i)_{i=1}^n$-*equilibrium* of $\Gamma$ is a distribution $\mu \in \Delta(\mathcal{X})$ such that

$$\mathbb{E}_{\boldsymbol{x} \sim \mu} \left[ u_i(\phi_i(\boldsymbol{x}_i), \boldsymbol{x}_{-i}) - u_i(\boldsymbol{x}) \right] \leq \epsilon$$

for all players $i$ and deviations $\phi_i \in \Phi_i$.

As discussed in Section 5, several special cases of $\Phi_i$ are well-studied and interesting:

- When $\Phi_i$ contains the set of constant functions and the projection $\pi_i$, the set of $(\Phi_i)_{i=1}^n$-equilibria are the CCEs.

- When $\Phi_i$ consists of linear functions depending only on $\boldsymbol{x}_i$, *i.e.*, functions of the form $\boldsymbol{x} \mapsto \mathbf{A}\boldsymbol{x}_i$ for matrices $\mathbf{A}$, $(\Phi_i)_{i=1}^n$-equilibria are LCEs, which correspond to CEs in the special case of normal-form games.

- When $\Phi_i$ consists of all functions $\mathcal{X} \to \mathcal{X}_i$, $(\Phi_i)_{i=1}^n$-equilibria are Nash equilibria.

Now let $\mathcal{X} = \mathcal{X}_1 \times \cdots \times \mathcal{X}_n \subset \mathbb{R}^d$ where $d = \sum_i d_i$, and define $\Phi \subseteq \mathcal{X}^{\mathcal{X}}$ to be the set of all functions of the form

$$\boldsymbol{x} \mapsto (\boldsymbol{x}_1, \dots, \phi_i(\boldsymbol{x}), \dots, \boldsymbol{x}_n)$$

for players $i$ and functions $\phi_i \in \Phi_i$. We will abuse notation and also call these functions $\phi_i : \mathcal{X} \to \mathcal{X}$. Moreover, let $F : \mathcal{X} \to \mathbb{R}^{\sum_i d_i}$ be given by $F(\boldsymbol{x}) = -(\nabla_{\boldsymbol{x}_1} u_1(\boldsymbol{x}), \dots, \nabla_{\boldsymbol{x}_n} u_n(\boldsymbol{x}))$.

**Proposition F.2.** *If $\mu$ is an $\epsilon$-approximate $\Phi$-EVI solution of $F$, then $\mu$ is an $\epsilon$-approximate $(\Phi_i)_{i=1}^n$-equilibrium $\Gamma$. The converse holds for $\epsilon = 0$.*

*Proof.* Suppose first that $\mu \in \Delta(\mathcal{X})$ is an $\epsilon$-approximate $\Phi$-EVI solution of $F$. Then, for any player $i$ and deviation $\phi_i \in \Phi_i$, we have

$$\mathbb{E}_{\boldsymbol{x} \sim \mu} \left[ u_i(\phi_i(\boldsymbol{x}), \boldsymbol{x}_{-i}) - u_i(\boldsymbol{x}) \right] \leq \mathbb{E}_{\boldsymbol{x} \sim \mu} \langle \nabla_{\boldsymbol{x}_i} u_i(\boldsymbol{x}), \phi_i(\boldsymbol{x}) - \boldsymbol{x}_i \rangle = \mathbb{E}_{\boldsymbol{x} \sim \mu} \langle -F(\boldsymbol{x}), \phi_i(\boldsymbol{x}) - \boldsymbol{x} \rangle \leq \epsilon,$$

where the first inequality is concavity and the second is the definition of $\Phi$-EVI. Conversely, suppose that $\mu$ is an (exact) $(\Phi_i)_{i=1}^n$-equilibrium of $\Gamma$. For $\lambda \in \mathbb{R}$ let $\phi_i^\lambda = \lambda \phi_i + (1 - \lambda)\pi_i$. Let $g : [0,1] \times \mathcal{X} \to \mathbb{R}$ be defined by

$$g(\lambda, \boldsymbol{x}) = u_i(\phi_i^\lambda(\boldsymbol{x}), \boldsymbol{x}_{-i}) - u_i(\boldsymbol{x}).$$

Then, $g$ is differentiable in $\lambda$ for any fixed $\boldsymbol{x}$, and $g$ is bounded. Let $G(\lambda) = \mathbb{E}_{\boldsymbol{x} \sim \mu} \, g(\lambda, \boldsymbol{x})$. Then $G(0) = 0$, and by the Leibniz rule, $G$ is differentiable with derivative

$$
\begin{aligned}
G'(0) &= \mathop{\mathbb{E}}_{\boldsymbol{x} \sim \mu} \nabla_\lambda g(0, \boldsymbol{x}) \\
&= \mathop{\mathbb{E}}_{\boldsymbol{x} \sim \mu} \left\langle \nabla_{\boldsymbol{x}_i} u_i(\boldsymbol{x}), \lim_{\lambda \to 0} \frac{1}{\lambda} \big( \phi_i^\lambda(\boldsymbol{x}) - \boldsymbol{x}_i \big) \right\rangle \\
&= \mathop{\mathbb{E}}_{\boldsymbol{x} \sim \mu} \langle \nabla_{\boldsymbol{x}_i} u_i(\boldsymbol{x}), \phi_i(\boldsymbol{x}) - \boldsymbol{x}_i \rangle \\
&= \mathop{\mathbb{E}}_{\boldsymbol{x} \sim \mu} \langle -F(\boldsymbol{x}), \phi_i(\boldsymbol{x}) - \boldsymbol{x} \rangle,
\end{aligned}
$$

where we use the chain rule, then the definition of $\phi_i^\lambda$, and finally the definition of $F$. But if $\mathbb{E}_{\boldsymbol{x} \sim \mu} \langle -F(\boldsymbol{x}), \phi_i(\boldsymbol{x}) - \boldsymbol{x} \rangle > 0$, then by definition of derivative, there is some $\lambda > 0$ for which $G(\lambda) > 0$, contradicting the definition of $(\Phi_i)_{i=1}^n$-equilibrium. $\qquad\square$

We now prove the following generalization of Proposition 5.3.

**Proposition F.3.** *For a given distribution $\mu \in \Delta(\mathcal{X})$, define the distribution $\mu^n \in \Delta(\mathcal{X}^n)$ by sampling $\boldsymbol{x} \sim \mu$ and outputting $(\boldsymbol{x}, \ldots, \boldsymbol{x}) \in \mathcal{X}^n$. Then the $(\Phi_1, \ldots, \Phi_n)$-equilibria of $\Gamma$ are precisely the $(\Phi, \ldots, \Phi)$-equilibria of $\Gamma^{\mathrm{sym}}$.*

*Proof.* $\mu^n$ is an $\epsilon$-approximate $(\Phi, \ldots, \Phi)$-equilibria of $\Gamma^{\mathrm{sym}}$ if and only if, for every player $i$ and linear map $\phi : \mathcal{X} \to \mathcal{X}$, we have

$$
\begin{aligned}
0 &\geq \frac{1}{n!} \sum_{\sigma \in \mathfrak{G}_n} \mathop{\mathbb{E}}_{\boldsymbol{x} \sim \mu} \big[ u_{\sigma(i)}(\boldsymbol{x}_1, \ldots, \phi(\boldsymbol{x})_{\sigma(i)}, \ldots, \boldsymbol{x}_n) - u_{\sigma^{-1}(i)}(\boldsymbol{x}) \big] \\
&= \frac{1}{n} \sum_{j \in [n]} \mathop{\mathbb{E}}_{\boldsymbol{x} \sim \mu} \big[ u_j(\boldsymbol{x}_1, \ldots, \phi(\boldsymbol{x})_j, \ldots, \boldsymbol{x}_n) - u_j(\boldsymbol{x}) \big].
\end{aligned}
$$

But this holds if and only if

$$
\mathop{\mathbb{E}}_{\boldsymbol{x} \sim \mu} \big[ u_j(\boldsymbol{x}_1, \ldots, \phi_j(\boldsymbol{x}), \ldots, \boldsymbol{x}_n) - u_j(\boldsymbol{x}) \big] \leq 0
$$

for every player $j$ and every $\phi_j \in \Phi_j$, which is precisely the definition of an $(\Phi_1, \ldots, \Phi_n)$-equilibria of $\Gamma$. $\qquad\square$

Proposition 5.3 follows by combining Proposition F.3 and Proposition F.2 in the special case when $\Phi_i = \Phi_{\mathsf{LIN}}(\mathcal{X}, \mathcal{X}_i)$.

### F.1. Anonymous linear correlated equilibria

For the special case where $\Phi_i = \Phi_{\mathsf{LIN}}(\mathcal{X}, \mathcal{X}_i)$, we have coined the resulting $(\Phi_i)_{i=1}^n$-equilibrium notion an *anonymous linear correlated equilibrium* (ALCE). We now compare ALCEs and LCEs in concave games. We now point out some intriguing properties of ALCEs, especially compared to LCEs and CEs.

In normal-form games $\Gamma$, LCEs and CEs coincide, and ALCEs lie strictly between LCEs and Nash equilibria, as can be seen in Figure 1. We now elaborate on the normal-form specific game-theoretic interpretation of ALCEs by giving an augmented game-based definition. For any fixed $\mu \in \Delta(\mathcal{X})$, consider the augmented game $\Gamma^\mu$ that proceeds as follows.

1. A correlation device samples $\boldsymbol{x} \sim \mu$.

2. Each player $i$ chooses a player $j$ (possibly not itself) and observes a sample $a_j \sim \boldsymbol{x}_j$, *independently from the samples of other players*. (In particular, if multiple players choose the same player $j$, then they get independent samples from $\boldsymbol{x}_j$).

3. Each player selects an action $a_i \in \mathcal{A}_i$ and gets utility $u_i(a_1, \ldots, a_n)$.

**Proposition F.4.** *A distribution $\mu \in \Delta(\mathcal{X})$ is a ALCE of $\Gamma$ if and only if the strategy profile in which every player requests an action for itself and then plays that action is a Nash equilibrium of $\Gamma^\mu$.*

The proof will use critically the following characterization of linear maps.

**Lemma F.5** (Fujii, 2023). *Let $\mathcal{X} = \mathcal{X}_1 \times \cdots \times \mathcal{X}_n$ where each $\mathcal{X}_i$ is a simplex $\mathcal{X}_i = \Delta(m_i)$. Then every linear map $\phi : \mathcal{X} \to \mathcal{X}_i$ is a convex combination of linear maps $\phi_j : \mathcal{X} \to \mathcal{X}_i$ that only depend on a single $\boldsymbol{x}_j$.*

*Proof of Proposition F.4.* Fix some $\mu \in \Delta(\mathcal{X})$ and suppose that it is not a ALCE, that is, there is some profitable deviation $\phi : \mathcal{X} \to \mathcal{X}_i$ for some player $i$. By Lemma F.5, it suffices to assume that $\phi$ only depends on one player's strategy $\boldsymbol{x}_j$. Moreover, a linear map $\phi : \mathcal{X}_j \to \mathcal{X}_i$ can be represented as $\boldsymbol{x}_j \mapsto \mathbf{A}\boldsymbol{x}_i$, where $\mathbf{A} \in \mathbb{R}^{m_i \times m_j}$ is column-stochastic. Again, it suffices to assume that $\phi$ is a vertex of the set of column-stochastic matrices, that is, $\mathbf{A}$ has exactly one 1 in each column. Now player $i$'s deviation benefit under deviation $\phi$ is given by

$$\mathbb{E}_{\boldsymbol{x}\sim\mu}[u_i(\phi_j(\boldsymbol{x}_j), \boldsymbol{x}_{-i}) - u_i(\boldsymbol{x})] = \mathbb{E}_{\substack{\boldsymbol{x}\sim\mu \\ a\sim\boldsymbol{x}}}\left[ \mathbb{E}_{a'_j\sim\boldsymbol{x}_j} u_i(\phi_j(a'_j), a_{-i}) - u_i(a) \right],$$

where the equality uses multilinearity of $a$. This is precisely the deviation benefit of the strategy in $\Gamma^\mu$ for player $i$ in which player $i$ chooses to sample $a'_j$ and then plays an action according to $\phi_j : [m_j] \to [m_i]$. The proposition now follows by observing that these are precisely the possible pure strategy deviations of player $i$ in $\Gamma^\mu$. $\qquad\square$

We make several more observations about the relationship between ALCEs and other notions of equilibrium in games.

- Proposition F.4 generalizes beyond normal-form games, but needs to be modified. For example, for (single-step) Bayesian games where each $\mathcal{X}_i$ is itself a product of simplices, it follows from a similar proof that, in the augmented game $\Gamma^\mu$, player $i$ should be allowed to observe its own type first, and then select both another player $j$ and a type $\theta_j$ of that player at which to ask for a recommendation. (Another way to see this is that the EVI formulation does not distinguish Bayesian games from their *agent form* (Kuhn, 1953), where each player-type pair is treated as a separate player.)

  Even more generally, for extensive-form games, we can generalize ALCEs using a characterization of the linear maps $\mathcal{X} \to \mathcal{X}_i$ due to Zhang et al. (2024b): in $\Gamma^\mu$, player $i$ first may observe its first recommendation at any time of its choosing, and may delay its choice of which player $j$ to observe until that point.

- In normal-form games, CEs can be without loss of generality defined as distributions over *pure* action profiles $\mathcal{A} = \mathcal{A}_1 \times \cdots \times \mathcal{A}_n$ instead of distributions over mixed strategy profiles $\mathcal{X} = \mathcal{X}_1 \times \cdots \times \mathcal{X}_n$ (Aumann, 1974). By *without loss of generality* here, we mean the following: given any $\mu \in \Delta(\mathcal{X})$, define $\mu' \in \Delta(\mathcal{A})$ by sampling $\boldsymbol{x} \sim \mu$, then $a_i \sim \boldsymbol{x}_i$ for each $i$. Then $\mu$ is a correlated equilibrium if and only if $\mu'$ is.

  This phenomenon is *not* true for ALCEs. Indeed, for two-player games, if $\mu' \in \Delta(\mathcal{A})$ is a ALCE, then in fact $\mu'$ is a distribution over pure Nash equilibria, which in general may not even exist! It is thus critical in our definition that $\mu$ be allowed to be a distribution over *mixed* strategy profiles, not just *pure* strategy profiles.

- We have shown that there is an efficient algorithm for computing *one* (approximate) ALCE. We leave as an open question the complexity of computing an *optimal* (*e.g.*, welfare-maximizing) ALCE (when the number of players $n$ is a constant). Optimal CEs can be computed efficiently in this setting, because the set of CEs $\mu \in \Delta(\mathcal{A})$ is bounded by a small number of linear constraints; however, this fails for ALCEs because, as above, we need to optimize over $\mu \in \Delta(\mathcal{X})$.

## G. Local $(\Phi_i)_{i=1}^n$-Equilibria in Nonconcave Games

This section connects $\Phi$-EVIs with a solution concept recently put forward by Cai et al. (2024a) (see also Şeref Ahunbay 2025) in the context of nonconcave games (Proposition G.4).

**Nonconcave games** Consider an $n$-player game in which each player $i \in [n]$ has a convex and compact strategy set $\mathcal{X}_i$, and a differentiable utility function $u_i : \mathcal{X}_1 \times \cdots \times \mathcal{X}_n \to \mathbb{R}$. Crucially, there is now no assumption that $u_i$ is concave. In this setting, our framework suggests the following definition.

**Definition G.1.** Given sets of functions $\Phi \subseteq \mathcal{X}_i^{\mathcal{X}_i}$, an $\epsilon$-*approximate local* $(\Phi_i)_{i=1}^n$-*equilibrium* in an $n$-player nonconcave game is a distribution $\mu \in \Delta(\mathcal{X}_1 \times \cdots \times \mathcal{X}_n)$ such that for any player $i \in [n]$ and deviation $\phi_i \in \Phi_i$,

$$\mathbb{E}_{\boldsymbol{x}\sim\mu} \langle \nabla_{\boldsymbol{x}_i} u_i(\boldsymbol{x}), \phi_i(\boldsymbol{x}_i) - \boldsymbol{x} \rangle \le \epsilon.$$

Theorem 4.1 immediately implies the following result when $\Phi_i = \Phi_{\mathsf{LIN}}(\mathcal{X}_i, \mathcal{X}_i)$; as before, in what follows, we assume a membership oracle for each $\mathcal{X}_i$.

**Corollary G.2.** *Suppose $\|\nabla u_i(\boldsymbol{x})\| \leq B$ for every player $i \in [n]$ and profile $\boldsymbol{x} \in \mathcal{X}_1 \times \cdots \times \mathcal{X}_n$. Then, there is a $\mathrm{poly}(d, \log(B/\epsilon))$-time algorithm that outputs an $\epsilon$-approximate local $(\Phi_i)_{i=1}^n$-equilibrium.*

Similarly, the existence of linear swap-regret minimizers for arbitrary polytopes $\mathcal{X}_i$ (Daskalakis et al., 2025) immediately implies the following.

**Corollary G.3.** *There is an independent no-regret learning algorithm that computes $\epsilon$-approximate local $(\Phi_i)_{i=1}^n$-equilibria in $\mathrm{poly}(d, 1/\epsilon)$ rounds and $\mathrm{poly}(d, 1/\epsilon)$ per-round runtime.*

Cai et al. (2024a) also studied the problem of computing local $(\Phi_i)_{i=1}^n$-equilibria in nonconcave games. They defined $\epsilon$-local $(\Phi_i)_{i=1}^n$-equilibria instead by restricting the magnitudes of the deviations to the "first-order" regime where local deviations cannot change the gradients by too much. In particular, they assume that utility functions $u_i$ are smooth, in the sense that

$$\|\nabla_{\boldsymbol{x}_i} u_i(\boldsymbol{x}_i, \boldsymbol{x}_{-i}) - \nabla_{\boldsymbol{x}_i} u_i(\boldsymbol{x}_i', \boldsymbol{x}_{-i})\|_2 \leq L\|\boldsymbol{x}_i - \boldsymbol{x}_i'\| \quad \forall \boldsymbol{x}_i, \boldsymbol{x}_i' \in \mathcal{X}_i, \forall \boldsymbol{x}_{-i} \in \bigtimes_{i' \neq i} \mathcal{X}_{i'},$$

where $L > 0$ is a constant. Then, they restrict deviations to only slightly perturb the strategies, that is, for a given set $\Phi_i \subseteq \mathcal{X}_i^{\mathcal{X}_i}$, they define a set

$$\Phi_i(\delta) := \{\lambda\phi_i + (1 - \lambda)\,\mathrm{Id} : \phi_i \in \Phi_i, \lambda \leq \delta/D_i\},$$

where $\mathrm{Id} : \mathcal{X} \to \mathcal{X}$ is the identity function and $D_i$ is the $\ell_2$-diameter of $\mathcal{X}_i$, *i.e.*, $\|\boldsymbol{x} - \boldsymbol{x}'\|_2 \leq D_i$ for all $\boldsymbol{x}, \boldsymbol{x}' \in \mathcal{X}_i$. With this restriction, they show (Cai et al., 2024a, Lemma 1 and Theorem 10) that $\Phi$-regret minimizers converge to $\Phi(\delta)$-equilibria, in the sense that

$$\mathbb{E}_{\boldsymbol{x} \sim \mu}[u_i(\phi_i(\boldsymbol{x}_i), \boldsymbol{x}_{-i}) - u_i(\boldsymbol{x})] \leq \frac{\delta}{D_i} \frac{\Phi\text{-Reg}_i^{(T)}}{T} + \frac{\delta^2 L}{2},$$

where $\Phi\text{-Reg}_i$ is the $\Phi_i$-regret of Player $i \in [n]$, for all players $i$ and deviations $\phi_i \in \Phi_i(\delta)$. Our results imply theirs, in the following sense.

**Proposition G.4.** *Any $\epsilon$-approximate local $(\Phi_i)_{i=1}^n$-equilibrium $\mu$ (per Definition G.1) satisfies*

$$\mathbb{E}_{\boldsymbol{x} \sim \mu}[u_i(\phi_i(\boldsymbol{x}_i), \boldsymbol{x}_{-i}) - u_i(\boldsymbol{x})] \leq \frac{\delta\epsilon}{D_i} + \frac{\delta^2 L}{2}$$

*for any player $i \in [n]$ and deviation $\phi_i \in \Phi_i(\delta)$.*

*Proof.* Write $\phi_i = \lambda\phi_i^* + (1 - \lambda)\,\mathrm{Id}$ for some $\phi_i^* \in \Phi_i$. Then,

$$u_i(\phi_i(\boldsymbol{x}_i), \boldsymbol{x}_{-i}) - u_i(\boldsymbol{x}) \leq \langle \nabla_{\boldsymbol{x}_i} u_i(\boldsymbol{x}), \phi_i(\boldsymbol{x}_i) - \boldsymbol{x}_i \rangle + \frac{L}{2}\|\phi_i(\boldsymbol{x}_i) - \boldsymbol{x}_i\|_2^2 \leq \frac{\delta}{D_i}\langle \nabla_{\boldsymbol{x}_i} u_i(\boldsymbol{x}), \phi_i^*(\boldsymbol{x}_i) - \boldsymbol{x}_i \rangle + \frac{\delta^2 L}{2},$$

where the last inequality uses the fact that $\lambda \leq \delta/D_i$ and therefore $\|\phi_i(\boldsymbol{x}_i) - \boldsymbol{x}_i\|_2 \leq \lambda\|\phi_i^*(\boldsymbol{x}_i) - \boldsymbol{x}_i\|_2 \leq \lambda D_i \leq \delta$. Taking expectations over $\mu$ and applying the definition of $\epsilon$-approximate local $(\Phi_i)_{i=1}^n$-equilibrium completes the proof. $\square$

However, our results improve on theirs in several ways:

- We believe that the formulation of local $(\Phi_i)_{i=1}^n$-equilibria using gradients directly instead of restricting to small perturbations is more natural and more directly conveys what it means for a distribution to be a local $(\Phi_i)_{i=1}^n$-equilibrium without introducing too many hyperparameters; one of the notions proposed by Şeref Ahunbay (2025, Definition 6) also shares this advantage.

- Our results do not require the smoothness of the utility functions $u_i$.

- We have an ellipsoid-based algorithm that computes local $(\Phi_i)_{i=1}^n$-equilibria with convergence rate depending on $\log(1/\epsilon)$, whereas no-regret algorithms only achieve $\mathrm{poly}(1/\epsilon)$ convergence rate.

- Although we do not explicitly state it here, Definition G.1 and Corollary G.2 extend directly to the case where $\Phi_i = \Phi_{\mathsf{LIN}}(\mathcal{X}, \mathcal{X}_i)$ (instead of $\Phi_{\mathsf{LIN}}(\mathcal{X}_i, \mathcal{X}_i)$). Per Appendix F, this can yield an even smaller set of equilibria.

