# OpenReview forum: "Expected Variational Inequalities"
_ICML.cc/2025/Conference — ICML 2025 oral_

### Official Review · Reviewer_7Vec · 2025-03-11

**Overall Recommendation:** 4

**Summary:**

This paper considers a relaxation of the variational inequality problem (VIP), the $\phi$-Expected Variational Inequality problem (EVI) (see Def. 1.2). This definition relaxes the 'classical' definition of the VIP in three ways: (1) the objective is to find a distribution $\mu$ such that the inequality holds in expectation, (2) the comparator is not wrt any $x \in X$, but with respect to $\phi(x)$ where $\phi:X \rightarrow X$ is a map (which is assumed to be linear for most results) and (3) the inequality is relaxed to hold wrt $-\epsilon$.
The authors provide complexity results (PPAD hardness for non-linear $\phi$), a polynomial bound if $\phi$ is linear (Thm 4.1), a regret bound and give application examples for games. Further, they give some conditions under which an $\epsilon$ solution for a EVI is also an $\epsilon$ solution for a VI.

**Claims And Evidence:**

This is theoretical work, all claims are supported by proofs.

**Essential References Not Discussed:**

To the best of my knowledge, all relevant references are discussed.

**Experimental Designs Or Analyses:**

No experiments are provided. (but this is fine since this is theoretical work)

**Methods And Evaluation Criteria:**

This is theoretical work, thus benchmarks etc do not apply.

**Other Comments Or Suggestions:**

none.

**Other Strengths And Weaknesses:**

The paper is well written. I like the versatility of the proofs and results.

 I am a bit unsure about the relevance of the solution concept of $\phi$-EVI. To the best of my understanding, if $\Phi$ contains all functions mapping into $X$, EVIs would correspond to VIs if $\mu$ is restricted to Dirac measures. However, the class of mappings $\Phi$ is quite restrictive. Thus I am not yet convinced of the relevance of the solution concept, but I may also miss something.  Thus I am happy to revise my evaluation.

Also see my questions.

**Questions For Authors:**

I am a bit confused regarding the interpretation of the solution for $\phi$-EVIs. For example, in the setting of games: Is there a clear relation to the $\phi$-equilibria considered in Cai et al.?
Is there a similar solution concept in optimization to which you can relate $\phi$-EVIs (as VIs relate to convex optimization)?

**Relation To Broader Scientific Literature:**

To the best of my understanding, the paper follows a similar idea as in Cai et. al. 2024 and (to some extent) generalizes this to VIP. Further, it falls into the brought area of 'local (cc) equilibria/first order optimality' for non-concave games and approaches this from the VIP perspective.

**Theoretical Claims:**

I did check some, but not all.

---

> ### Author Rebuttal · Authors · 2025-04-01
>
> We thank the reviewer for their feedback and service.
>
> *"To the best of my understanding, if  $\Phi$ contains all functions [...] EVIs would correspond to VIs if $\mu$  is restricted to Dirac measures."*
>
> We clarify that this equivalence holds without any restriction on $\mu$; it doesn’t have to be a Dirac measure.
>
> *"Thus I am not yet convinced of the relevance of the solution concept"*
>
> Our main positive result captures the setting where $\Phi$ contains all linear functions, which we argue is a strong solution concept. Even in the special case of normal-form games, it is tighter (that is, closer to Nash equilibria) than the usual notion of a correlated equilibrium (see Figure 1). Moreover, our hardness results preclude efficient computation when $\Phi$ contains even quadratic functions; as a result, there is a fundamental sense in which our solution concept is the best one can hope for using efficient algorithms.
>
> *"In the setting of games: Is there a clear relation to the $\Phi$-equilibria considered in Cai et al.?"*
>
> In the special case of nonconcave games, our solution captures the one considered by Cai et al., as we show and discuss in Appendix G. In fact, when $\Phi$ contains all linear functions, our solution concept is tighter (see Figure 1). We also argue that our definition is more natural and simple, while also applying to any variational inequality problem. Besides those points, we provide significant algorithmic improvements compared to Cai et al., as we point out in Appendix G.
>
> *"Is there a similar solution concept in optimization to which you can relate?"*
>
> To the best of our knowledge, there is no existing solution concept in optimization that directly relates to $\Phi$-EVIs, except in more specific classes of problems (see Section 6). Given how natural this concept is, we believe that introducing it to the optimization community is an important contribution.

---

### Official Review · Reviewer_GSQj · 2025-03-13

**Overall Recommendation:** 5

**Summary:**

The authors relax VI to a new notion of EVI. They then take a game-theoretic approach to design polynomial-time approximation algorithms for EVIs.

**Claims And Evidence:**

Yes.

**Essential References Not Discussed:**

N/A

**Experimental Designs Or Analyses:**

N/A

**Methods And Evaluation Criteria:**

Yes.

**Other Comments Or Suggestions:**

It would be helpful if the authors could compare the strength of VI to EVI in terms of modeling power. For example, are there still many VI problems for which EVI suffices? Furthermore, are there any problems where EVI is actually more natural of a model than VI?

**Other Strengths And Weaknesses:**

The paper seems well written and not only introduces a reasonable relaxation of a known useful framework, but also presents provable approximation algorithms for the relaxed problem. The only weakness I can see is whether EVI is too great of a relaxation. Considering it brings a known hard problem down to polynomial time complexity, there must be some loss in modeling power or guarantees that result. Either way, I believe the insights are still meaningful even if EVI is too much of a relaxation.

**Questions For Authors:**

See my suggestions above.

**Relation To Broader Scientific Literature:**

VIs are a general way to capture many important problems across different fields. This paper addresses the complexity issues of VI problems by introducing a new notion that retains the key flavor of VI while making them computationally tractable. As such this work has the potential to lead to further insights to make VI problems more efficiently solvable, and out of the box gives efficient methods that could be promising heuristics to known VI problems. Overall, I believe this paper has the potential to have significant impact on the literature.

**Theoretical Claims:**

Yes.

---

> ### Author Rebuttal · Authors · 2025-04-01
>
> We thank the reviewer for their feedback and service.
>
> *“It would be helpful if the authors could compare the strength of VI to EVI in terms of modeling power. For example, are there still many VI problems for which EVI suffices? Furthermore, are there any problems where EVI is actually more natural of a model than VI?”*
>
> This is a great question. If we look at the special case of games, $\Phi$-EVIs (which subsume correlated equilibria) are in many ways more natural than VIs (that is, Nash equilibria). Indeed, they often lead to better outcomes in terms of welfare as they enable correlation between the players, which in many applications is more reasonable than assuming that players are acting independently (a traffic light is a common example of a correlation device). This shows that there are problems in which EVIs are, in many ways, more natural than VIs. Furthermore, given how expressive games are from an optimization standpoint, we argue that this already makes a compelling case for EVIs compared to VIs. That being said, we believe that further understanding the power of EVIs and how they compare against VIs is an important question for follow-up work.

---

### Official Review · Reviewer_UNCg · 2025-03-13

**Overall Recommendation:** 4

**Summary:**

The paper studies the Variational Inequality (VI) problem, a well-known problem used in various optimisation problems, often in settings where equilibria need to be computed. The general problem is intractable, so there is a wide literature identifying subclasses of the VI problem for which it is tractable.

This paper proposes an alternative approach: relaxing the definition of the solution to the problem. Instead of requiring a single point in a convex space that should satisfy the central variational inequality, the *expected* variational inequality (EVI) problem asks to find a *distribution* of points that *in expectation* satisfy the inequality.

This problem is parametrised by the size of the set of deviations ($\Phi$) from the convex space for which the inequality must hold. The larger the set of deviations, the smaller the found distribution will be.

The paper considers different types of maps $\Phi$, including constant and linear map. A key contribution is a complexity result for when $\Phi$ contains only linear maps. In this case, the $\Phi$-EVI problem can be solved in time polynomial in dimension $d$ of the problem space and logarithm of the inverse of the precision parameter $\epsilon$. Since this result relies on the ellipsoid against hope (EAH) algorithm, which is slow in practice, the paper proposes an alternative algorithm that improves upon the state of the art in terms of the per-iteration complexity.

Furthermore, the paper presents equivalence and performance results, and identifies applications of (constant-$\Phi$) EVIs.

**Claims And Evidence:**

I honestly do not have the background to be an accurate judge of this. I am unfamiliar with many of the concepts discussed in this paper, and Wikipedia can only get you so far. I have no obvious reasons to doubt any of the claims that are made.

**Essential References Not Discussed:**

I do not consider myself knowledgable enough to judge this.

**Experimental Designs Or Analyses:**

N/A

**Methods And Evaluation Criteria:**

The main claims are theoretical and are supported by proofs in the appendix. That seems appropriate to me.

**Other Comments Or Suggestions:**

- The bibliography is a bit sloppy and inconsistent. For some entries the full name of the venue is given, for others just an abbreviation. Many entries seem to be missing page numbers.

**Other Strengths And Weaknesses:**

I really appreciate the efforts to make the paper somewhat standalone by including a recap of the ellipsoids against hope (EAH) algorithm. I also find that the paper is well-written in terms of structure and signposting. The writing is clean, with hardly any typos. The paragraph labelling and subsection labelling is inconsistent and confusing to me.

The main research direction and contribution seem valuable and interesting to me. I also like the efforts made by the authors to connect their results to different fields, and their discussion of their finding's implications for game-theoretic questions.

I personally dislike the writing style. Take the second sentence of the introduction. It spans 14 lines: almost a quarter page. I would've preferred that to be split up in a few sentences, organised by application category or something. Elsewhere in the paper there's also needlessly lavish prose. Why write "in lieu of" when "instead of" is right there? Going for the simpler option would even have saved the authors an `{\it }`. I personally believe that scientific writing should be as accessible as possible. Hence, I wish that the authors would have kept the language more simple.

**Questions For Authors:**

I do not feel like I have the expertise to formulate questions whose answers would make a difference in my judgement.

**Relation To Broader Scientific Literature:**

To a layperson like me, at least, it seems like the paper positions the main contributions well in relation to the existing literature. In doing so, it covers applications and theoretical results spanning several decades. The appendix contains more details on the relationship between EVIs and different game theory concepts.

**Theoretical Claims:**

I read through some of the proofs, but do not have the expertise to confidently assess their correctness.

---

> ### Author Rebuttal · Authors · 2025-04-01
>
> We thank the reviewer for their feedback and service.
>
> We are happy to adjust the writing style and follow any other suggestions for the revision. Regarding the bibliography, we follow the convention that page numbers are omitted for conference publications, which explains why some references have page numbers and others don’t. We are happy to change that if the reviewer finds it inconsistent.

---

### Decision · Program_Chairs · 2025-05-01

**Decision:**

Accept (oral)

**Comment:**

Summary: The paper introduces and analyzes a natural relaxation of VIs, namely expected variational inequalities (EVIs), where the goal is to find a distribution that satisfies the VI constraint in expectation. By adapting recent techniques from game theory, it shows that, unlike VIs, EVIs can be solved in polynomial time under general (nonmonotone) operators.

On Reviews: All reviewers really enjoy reading this paper. It is well written with clear contributions. Excellent work from the authors.

The ideas presented in the paper are novel and worth exploring further in the future, especially their impact on a wide range of applications.
I advise the authors to incorporate the feedback they received into the updated version of their work. I recommend acceptance.